# Semialgebraic Representation of Monotone Deep Equilibrium Models and Applications to Certification

**Tong Chen**
LAAS-CNRS
Université de Toulouse
31400 Toulouse, France
tchen@laas.fr

**Jean-Bernard Lasserre**
LAAS-CNRS & IMT
Université de Toulouse
31400 Toulouse, France
lasserre@laas.fr

**Victor Magron**
LAAS-CNRS
Université de Toulouse
31400 Toulouse, France
vmagron@laas.fr

**Edouard Pauwels**
IRIT & IMT
Université de Toulouse
31400 Toulouse, France
edouard.pauwels@irit.fr

## Abstract

Deep equilibrium models are based on implicitly defined functional relations and have shown competitive performance compared with the traditional deep networks. Monotone operator equilibrium networks (monDEQ) retain interesting performance with additional theoretical guaranties. Existing certification tools for classical deep networks cannot directly be applied to monDEQs for which much fewer tools exist. We introduce a semialgebraic representation for ReLU based monDEQs which allows to approximate the corresponding input output relation by semidefinite programming (SDP). We present several applications to network certification and obtain SDP models for the following problems : robustness certification, Lipschitz constant estimation, ellipsoidal uncertainty propagation. We use these models to certify robustness of monDEQs w.r.t. a general $L_q$ norm. Experimental results show that the proposed models outperform existing approaches for monDEQ certification. Furthermore, our investigations suggest that monDEQs are much more robust to $L_2$ perturbations than $L_\infty$ perturbations.

## 1   Introduction

With the increasing success of Deep Neural Networks (DNN) (e.g. computer vision, natural language processing), one witnesses a significant increase in size and complexity (topology and activation functions). This generates difficulties for theoretical analysis and a posteriori performance evaluation. This is problematic for applications where robustness issues are crucial, for example inverse problems (IP) in scientific computing. Indeed such IPs are notoriously ill-posed and as stressed in the March 2021 issue of SIAM News [1], *"Yet DL has an Achilles' heel. Current implementations can be highly unstable, meaning that a certain small perturbation to the input of a trained neural network can cause substantial change in its output. This phenomenon is both a nuisance and a major concern for the safety and robustness of DL-based systems in critical applications—like healthcare—where reliable computations are essential"*. Indeed, the Instability Theorem [1] predicts unavoidable lower bound on Lipschitz contants, which may explain the lack of stability of some DNNs, over-performing on training sets. This underlines the need to evaluate precisely a posteriori critical indicators, such as Lipschitz constants of DNNs. However, obtaining an accurate upper bounds on the Lipschitz constant of a DNN is a hard problem, it reduces to *prove a global* inequality "$\Psi(\mathbf{x}) \geq 0$ for all $\mathbf{x}$ in a domain",

35th Conference on Neural Information Processing Systems (NeurIPS 2021).

i.e., to provide a *proof of positivity* for a function $\Psi$, which has no simple explicit expression. Even for modest size DNNs this task is practically challenging, previous successful attempts [8, 28] were restricted in practice to no more than two hidden layers with less than a hundred nodes. More broadly existing attempts to DNN certification rely either on zonotope calculus, linear programming (LP) or hierarchies of SDP based on positivity certificates from algebraic geometry [34], which may suffer from the curse of dimensionality.

Recently, implicit deep learning [12] arises as a generalization of the recursive rules of traditional feedforward neural networks. Specifically, *Deep Equilibrium Models (DEQ)* [2] have emerged as a potential alternative to classical DNNs. With their much simpler layer structure, they provide competitive results on machine learning benchmarks [2, 3]. Training DEQs involves solving fix-point equations, which requires further conditions to make the iteration converge. Fortunately, *Monotone operator equilibrium network (monDEQ)* introduced in [45] satisfies such conditions. Moreover, the authors in [33] provide explicit bounds on global Lipschitz constant of monDEQs (w.r.t. the $L_2$-norm) which can be used for robustness certification.

From a certification point of view, DEQs have the definite advantage of having only one implicit layer compared to DNNs and therefore is potentially more amenable to sophisticated techniques (e.g. algebraic certificates of positivity) which rapidly face their limit even for classical DNNs with modest size (but long depth). Therefore monDEQs constitute a class of DNNs for which robustness certification, uncertainty propagation or Lipschicity could potentially be investigated in a more satisfactory way than classical networks. Contrary to DNNs, for which a variety of tools have been developed, certification of DEQ modeld is relatively open, the only available tool is the Lipschitz bound in [33].

**Contribution**

We present three general semialgebraic models of ReLU monDEQ for certification ($p \in \mathbb{Z}_+ \cup \{+\infty\}$):

- *Robustness Model* for network $L_p$ robustness certification.

- *Lipschitz Model* for network Lipschitz constant estimation with respect to any $L_p$ norm.

- *Ellipsoid Model* for ellipsoidal outer-approximation of the image by the network of a polyhedra or an ellipsoid.

All these models can be used for robustness certification, a common task which we consider experimentally. And the main originality of this work is to successfully apply the proposed techniques to monDEQ networks (especially for the Ellipsoid model which is a novel form of reachability analysis), which was not proposed before. Both Lipschitz and Ellipsoid models can in addition be used for further a posteriori analyses. Interestingly, all three models are given by solutions of semidefinite programs (SDP), obtained by Shor relaxation of a common semialgebraic representation of ReLU monDEQs. Our models are all evaluated to simple ReLU monDEQs on MNIST dataset similar as [45, 33] on the task of robustness certification. We demonstrate that all three models ourperform the approach of [33] and the Robustness Model being the most efficient. Our experiments also suggest that DEQs are much less robust to $L^\infty$ perturbations than $L^2$ perturbations, in contrast with classical DNNs [36] which have been shown to be robust on MNIST dataset to the level we are considering ($\varepsilon = 0.1$).

**Related works**

Neural network certification is a challenging topic in machine learning, contributions include:

**Robustness certification of DNNs**  Even with high test accuracy, DNNs are very sensitive to tiny input perturbations, see e.g. [42, 16]. Robustness to input perturbation has been investigated in many different works with various techniques, including SDP relaxation in SDP-cert [36], first-order dual SDP method [10], abstract interpretation with ERAN [15, 41], multi-neuron convex relaxations with PRIMA [31], GPU-based method GPUPoly [30] which can scale to large networks with one million neurons and 34 layers, scalable quantitative verification framework with PROVERO [4], LP relaxation in Reluplex [25], analytical certification with Fast-lin [44] and CROWN [47], and their extension to convolutional neural networks with CNN-Cert [7]. All these methods are restricted to DNNs or CNNs and do not directly apply to DEQs.

**Lipschitz constant estimation of DNNs**   Lipschitz constant of DNNs can be used for robustness certification [42, 18]. Existing contributions include naive layer-wise product bound [22, 43], controlled ordinary differential equations (ODE) method [20], zonotope-based method [24] which can be further extended to generative models, Mixed-Integer Programming (MIP) [23] for exactly computing the local Lipschitz constant of ReLU networks, the LP relaxations for CNNs [48] and DNNs [28], as well as SDP relaxations [35, 14, 8].

**Lipschitz constant estimation of DEQs**   The authors in [33] provide an optimization-free upper bound of the Lipschitz constant of monDEQs depending on network weights. This bound is valid for $L^2$ norm and we present a general model for arbitrary $L^p$ norm. Based on the works of [33], the authors in [38] introduced new parameterizations of DEQs that admit a Lipschitz bound during training via unconstrained optimization.

## 2   Preliminary Background and Notations

We consider the *monotone operator equilibrium network (monDEQ)* [45], with a single implicit hidden layer. The main difference between monDEQ and *deep equilibrium network (DEQ)* [2] is that *strong monotonicity* is enforced on weight matrix and activation functions to guarantee the convergence of fixed point iterations. The authors in [45] proposed various structures of implicit layer, we only consider fully-connected layers, investigation of more advanced convolutional layers is in our list of future works.

**Network description**: Denote by $F : \mathbb{R}^{p_0} \to \mathbb{R}^K$ a fully-connected monDEQ for classification, where $p_0$ is the input dimension and $K$ is the number of labels. Let $\mathbf{x}_0 \in \mathbb{R}^{p_0}$ be the input variable and $\mathbf{z} \in \mathbb{R}^p$ be the variable in the implicit layer. We consider the ReLU activation function, which is simply defined as $\mathrm{ReLU}(x) = \max\{0, x\}$, and the output of the monDEQs can be written as

$$F(\mathbf{x}_0) = \mathbf{C}\mathbf{z} + \mathbf{c}, \mathbf{z} = \mathrm{ReLU}(\mathbf{W}\mathbf{z} + \mathbf{U}\mathbf{x}_0 + \mathbf{u}), \qquad \text{(monDEQ)}$$

where $\mathbf{W} \in \mathbb{R}^{p \times p}, \mathbf{U} \in \mathbb{R}^{p \times p_0}, \mathbf{u} \in \mathbb{R}^p, \mathbf{C} \in \mathbb{R}^{K \times p}, \mathbf{c} \in \mathbb{R}^K$ are parameters of the network. The vector-valued function $F(\mathbf{x}_0)$ provides a score for each label $i \in \{1, \ldots, K\}$ associated to the input $\mathbf{x}_0$, the prediction corresponds to the highest score, i.e., $y_{\mathbf{x}_0} = \arg\max_{i=1,\ldots,K} F(\mathbf{x}_0)_i$. As in [45], the matrix $\mathbf{I}_p - \mathbf{W}$ is *strongly monotone*: there is a known $m > 0$ such that $\mathbf{I}_p - \mathbf{W} \succeq m\mathbf{I}_p$, this constraint can be enforced by specific parametrization of the matrix $\mathbf{W}$. With the monotonicity assumption, the solution to equation $\mathbf{z} = \mathrm{ReLU}(\mathbf{W}\mathbf{z} + \mathbf{U}\mathbf{x}_0 + \mathbf{u})$ is unique and can be evaluated using convergent algorithms, see [45] for more details.

**Robustness of monDEQs**: Given an input $\mathbf{x}_0 \in \mathbb{R}^{p_0}$, a norm $\|\cdot\|$, and a network $F : \mathbb{R}^{p_0} \to \mathbb{R}^K$ , let $y_0$ be the label of input $\mathbf{x}_0$, i.e., $y_0 = \arg\max_{i=1,\ldots,K} F(\mathbf{x}_0)_i$. For $\varepsilon > 0$, denote by $\mathcal{E} = \mathbb{B}(\mathbf{x}_0, \varepsilon, \|\cdot\|)$ the ball centered at $\mathbf{x}_0$ with radius $\varepsilon$ for norm $\|\cdot\|$. If for all inputs $\mathbf{x} \in \mathcal{E}$, the label of $\mathbf{x}$ equals $y_0$, i.e., $y = \arg\max_{i=1,\ldots,K} F(\mathbf{x})_i = y_0$, then we say that the network $F$ is $\varepsilon$-*robust* at input $\mathbf{x}_0$ for norm $\|\cdot\|$. An equivalent way to verify whether the network $F$ is $\varepsilon$-robust is to check that for all labels $i \neq y_0$, $F(\mathbf{x})_i - F(\mathbf{x})_{y_0} < 0$.

**Semialgebraicity of ReLU function**: The key reason why neural networks with ReLU activation function can be tackled using polynomial optimization techniques is *semialgebraicity* of the ReLU function, i.e., it can be expressed with a system of polynomial (in)equalities. For $x, y \in \mathbb{R}$, we have $y = \mathrm{ReLU}(x) = \max\{0, x\}$ if and only if $y(y - x) = 0, y \geq x, y \geq 0$. For $\mathbf{x}, \mathbf{y} \in \mathbb{R}^n$, we denote by $\mathrm{ReLU}(\mathbf{x})$ the coordinate-wise evaluation of ReLU function, and by $\mathbf{xy}$ the coordinate-wise product of $\mathbf{x}$ and $\mathbf{y}$. A subset of $\mathbb{R}^n$ defined by a finite conjunction of polynomial (in)equalities is called a *basic closed semialgebraic set*. The graph of the ReLU function is a basic closed semialgebraic set.

Going back to equation (monDEQ), we have the following equivalence:

$$\mathbf{z} = \mathrm{ReLU}(\mathbf{W}\mathbf{z} + \mathbf{U}\mathbf{x}_0 + \mathbf{u}) \Leftrightarrow \mathbf{z}(\mathbf{z} - \mathbf{W}\mathbf{z} - \mathbf{U}\mathbf{x}_0 - \mathbf{u}) = 0, \mathbf{z} \geq \mathbf{W}\mathbf{z} + \mathbf{U}\mathbf{x}_0 + \mathbf{u}, \mathbf{z} \geq 0, \quad (1)$$

where the right hand side is a system of polynomial (in)equalities. For the rest of the paper, mention of the ReLU function will refer to the equivalent polynomial system in (1).

**POP and Putinar's positivity certificate**: In general, a *polynomial optimization problem (POP)* has the form

$$\rho = \max_{\mathbf{x} \in \mathbb{R}^n} \{f(\mathbf{x}) : g_i(\mathbf{x}) \geq 0, i = 1, \ldots, p\}, \qquad \text{(POP}_0\text{)}$$

where $f$ and $g_i$ are polynomials whose degree is denoted by $\deg$. The robustness certification model (3.1), Lipischitz constant model (3.2) and ellipsoid model (3.3) are all POPs.

In most cases, the POPs are non-linear and non-convex problems, which makes them NP-hard. A typical approach to reduce the complexity of these problems is replacing the positivity constraints by Putinar's positivity certificate [34]. The problem (POP$_0$) is equivalent to

$$\rho = \min_{\lambda \in \mathbb{R}} \{\lambda : \lambda - f(\mathbf{x}) \geq 0, g_i(\mathbf{x}) \geq 0, i = 1, \ldots, p, \forall \mathbf{x} \in \mathbb{R}^n\}. \tag{POP}$$

In order to reduce the size of the feasible set of problem (POP), we replace the positivity constraint $\lambda - f(\mathbf{x}) \geq 0$ by a weighted *sum-of-square (SOS)* polynomial decomposition, involving the polynomials $g_i$. Let $d$ be a non-negative integer. Denote by $\sigma_0^d(\mathbf{x}), \sigma_i^d(\mathbf{x})$ some SOS polynomials of degree at most $2d$, for each $i = 1, \ldots, p$. Note that if $d = 0$, such polynomials are non-negative real numbers. Then the positivity of $\lambda - f(\mathbf{x})$ is implied by the following decomposition

$$\lambda - f(\mathbf{x}) = \sigma_0^d(\mathbf{x}) + \sum_{i=1}^{p} \sigma_i^{d-\omega_i}(\mathbf{x})g_i(\mathbf{x}), \quad \omega_i = \lceil \deg g_i/2 \rceil, \quad \forall \mathbf{x} \in \mathbb{R}^n, \tag{2}$$

for any $d \geq \max_i \omega_i$. Equation (2) is called the *order-$d$ Putinar's certificate*. By replacing the positivity constraint $f(\mathbf{x}) - \lambda \geq 0$ in problem (POP) by Putinar's certificate (2), we have for $d \geq \max_i \omega_i$,

$$\rho_d = \min_{\lambda \in \mathbb{R}} \{\lambda : \lambda - f(\mathbf{x}) = \sigma_0^d(\mathbf{x}) + \sum_{i=1}^{p} \sigma_i^{d-\omega_i}(\mathbf{x})g_i(\mathbf{x}), \, \omega_i = \lceil \deg g_i/2 \rceil, \forall \mathbf{x} \in \mathbb{R}^n\}. \tag{POP-$d$}$$

It is obvious that $\rho_d \geq \rho$ for all $d \geq \max_i \omega_i$. Under certain conditions (slightly stronger than compactness of the set of constraints), it is shown that $\lim_{d \to \infty} \rho_d = \rho$ [27]. The main advantage of relaxing problem (POP) to (POP-$d$) is that problem (POP-$d$) can be efficiently solved by *semidefinite programming (SDP)*. Indeed a polynomial $f$ of degree at most $2d$ is SOS if and only if there exists a *positive semidefinte (PSD)* matrix $\mathbf{M}$ (called a *Gram matrix*) such that $f(\mathbf{x}) = \mathbf{v}(\mathbf{x})^T \mathbf{M} \mathbf{v}(\mathbf{x})$, for all $\mathbf{x} \in \mathbb{R}^p$, where $\mathbf{v}(\mathbf{x})$ is the vector of monomials of degree at most $d$.

Problem (POP-$d$) is also called the *order-$d$ Lasserre's relaxation*. When the input polynomials are quadratic, the order-1 Lasserre's relaxation is also known as *Shor's relaxation* [40]. All our models are obtained using variations of Shor's relaxation applied to different POPs, see Section 3.3 for more details.

## 3 Semialgebraic Models for Certifying Robustness of Neural Networks

In this section, we introduce several models for certification of monDEQs. All the models are based on semialgebraicity of ReLU and $\partial$ReLU (the *subgradient* of ReLU, see Section 3.2) to translate our targeted problems to POPs. Then we use Putinar's certificates, defined in Section 2 to relax the non-convex problems to convex SDPs which can be solved efficiently using modern solvers.

**Notation**: Throughout this section, we consider a monDEQ for classification, denoted by $F$, with fixed, given parameters, $\mathbf{W} \in \mathbb{R}^{p \times p}, \mathbf{U} \in \mathbb{R}^{p \times p_0}, \mathbf{u} \in \mathbb{R}^p, \mathbf{C} \in \mathbb{R}^{K \times p}, \mathbf{c} \in \mathbb{R}^K$, where $p_0$ is the number of input neurons, $p$ is the number of hidden neurons, and $K$ is the number of labels. For $q \in \mathbb{Z}_+ \cup \{+\infty\}$, $\|\cdot\|_q$ is the $L_q$ norm defined by $\|\mathbf{x}\|_q := (\sum_{i=1}^{p_0} |x_i|^q)^{1/q}$ for all $\mathbf{x} \in \mathbb{R}^{p_0}$. Throughout this section $\epsilon > 0$ and $\mathbf{x}_0 \in \mathbb{R}^{p_0}$ are fixed, we denote by $\mathcal{E} := \mathbb{B}(\mathbf{x}_0, \varepsilon, \|\cdot\|_q) = \{\mathbf{x} \in \mathbb{R}^{p_0} : \|\mathbf{x} - \mathbf{x}_0\|_q \leq \varepsilon\}$ the ball centered at $\mathbf{x}_0$ with radius $\varepsilon$ for $L_q$ norm, a perturbation region. If $q < +\infty$, i.e., $q$ is a positive integer, $\|\mathbf{x} - \mathbf{x}_0\|_q \leq \varepsilon$ is equivalent to the polynomial inequality $\|\mathbf{x} - \mathbf{x}_0\|_q^q \leq \varepsilon^q$; if $q = \infty$, $\|\mathbf{x} - \mathbf{x}_0\|_q \leq \varepsilon$ is equivalent to $|\mathbf{x} - \mathbf{x}_0|^2 \leq \epsilon^2$ (where $|\mathbf{x}|$ denotes the vector of absolute values of coordinates of $\mathbf{x}$) which is a system of $p_0$ polynomial inequalities. Hence the input set $\mathcal{E}$ is a semialgebraic set for all considered $L_q$ norms. For a matrix $\mathbf{A} \in \mathbb{R}^{m \times n}$, its *operator norm* induced by the norm $\|\cdot\|$ is given by $\|\mathbf{A}\| := \inf\{\lambda : \|\mathbf{A}\mathbf{x}\| \leq \lambda\|\mathbf{x}\|, \forall \mathbf{x} \in \mathbb{R}^n\}$.

### 3.1 Robustness Model

Let $y_0$ be the label of $\mathbf{x}_0$ and let $\mathbf{z} \in \mathbb{R}^p$ be the variables in the monDEQ implicit layer. The proposed model directly estimates upper bounds on the gap between the score of label $y_0$ and the score of labels

different from $y_0$. Precisely, for $i \in \{1, \ldots, K\}$ such that $i \neq y_0$, denote by $\xi_i = (\mathbf{C}_{i,:} - \mathbf{C}_{y_0,:})^T$. For $\mathbf{x} \in \mathcal{E}$, the gap between its score of label $i$ and label $y_0$ is $F(\mathbf{x})_i - F(\mathbf{x})_{y_0} = \xi_i^T \mathbf{z}$. The Robustness Model for monDEQ reads:

$$\delta_i := \max_{\mathbf{x} \in \mathbb{R}^{p_0}, \mathbf{z} \in \mathbb{R}^p} \{\xi_i^T \mathbf{z} : \mathbf{z} = \text{ReLU}(\mathbf{Wz} + \mathbf{Ux} + \mathbf{u}), \mathbf{x} \in \mathcal{E}\}. \qquad \text{(CertMON-}i\text{)}$$

Using the semialgebraicity of both ReLU in (1), and set $\mathcal{E}$, problem (CertMON-$i$) is a POP for all $i$. As discussed in Section 2, one is able to derive a sequence of SDPs (Lasserre's relaxation) to obtain a converging serie of upper bounds of the optimal solution of (CertMON-$i$). For Robustness Model, we consider only the order-1 Lasserre's relaxation (Shor's relaxation), and denote by $\tilde{\delta}_i$ the upper bound of $\delta_i$ by Shor's relaxation, i.e., $\delta_i \leq \tilde{\delta}_i$. Recall that if for all label $i$ different from $y_0$, we have $F(\mathbf{x})_i < F(\mathbf{x})_{y_0}$, then the label of $\mathbf{x}$ is still $y_0$. This justifies the following claim:

**Certification criterion**: If $\tilde{\delta}_i < 0$ for all $i \neq y_0$, then the network $F$ is $\varepsilon$-robust at $\mathbf{x}_0$.

Robustness Model for DNNs has already been investigated in [36], where the authors also use Shor's relaxation as we do. Different from DNNs, we only have one implicit layer in monDEQ. Therefore, the number of variables in problem (CertMON-$i$) only depends on the number of input neurons $p_0$ and hidden neurons $p$.

## 3.2 Lipschitz Model

We bound the Lipschitz constant of monDEQ with respect to input perturbation. Recall that the *Lipschitz constant* of the vector-valued function $F$ (resp. $\mathbf{z}$) w.r.t. the $L_q$ norm and input ball $\mathcal{S} \supset \mathcal{E}$, denoted by $L_{F,\mathcal{S}}^q$ (resp. $L_{\mathbf{z},\mathcal{S}}^q$), is the smallest value of $L$ such that $\|F(\mathbf{x}) - F(\mathbf{y})\|_q \leq L\|\mathbf{x} - \mathbf{y}\|_q$ (resp. $\|\mathbf{z}(\mathbf{x}) - \mathbf{z}(\mathbf{y})\|_q \leq L\|\mathbf{x} - \mathbf{y}\|_q$) for all $\mathbf{x}, \mathbf{y} \in \mathcal{S}$. For $\mathbf{x}, \mathbf{x}_0 \in \mathcal{S}$, with $\|\mathbf{x} - \mathbf{x}_0\|_q \leq \epsilon$, we can estimate the perturbation of the output as follows:

$$\|F(\mathbf{x}) - F(\mathbf{x}_0)\|_q \leq L_{F,\mathcal{S}}^q \cdot \|\mathbf{x} - \mathbf{x}_0\|_q \leq \varepsilon L_{F,\mathcal{S}}^q, \qquad (3)$$

$$\|F(\mathbf{x}) - F(\mathbf{x}_0)\|_q \leq \|\|\mathbf{C}\|\|_q \cdot L_{\mathbf{z},\mathcal{S}}^q \cdot \|\mathbf{x} - \mathbf{x}_0\|_q \leq \varepsilon \|\|\mathbf{C}\|\|_q \cdot L_{\mathbf{z},\mathcal{S}}^q. \qquad (4)$$

The authors in [33] use inequality (4) with $q = 2$, as they provide an upper bound of $L_{\mathbf{z},\mathcal{S}}^2$. In contrast, our model provides upper bounds on Lipschitz constants of $F$ or $\mathbf{z}$ for arbitrary $L_q$ norms. We directly focus on estimating the value of $L_{F,\mathcal{S}}^q$ instead of $L_{\mathbf{z},\mathcal{S}}^q$.

Since the ReLU function is non-smooth, we define its *subgradient*, denoted by $\partial\text{ReLU}$, as the set-valued map $\partial\text{ReLU}(x) = 0$ for $x < 0$, $\partial\text{ReLU}(x) = 1$ for $x > 0$, and $\partial\text{ReLU}(x) = [0, 1]$ for $x = 0$. Similar to ReLU function, $\partial\text{ReLU}$ is also semialgebraic.

**Semialgebraicity of $\partial\text{ReLU}$**: If $x, y \in \mathbb{R}$, we have $y \in \partial\text{ReLU}(x)$, if and only if $y(y-1) \leq 0$, $xy \geq 0$, $x(y-1) \geq 0$. If $\mathbf{x}, \mathbf{y} \in \mathbb{R}^n$, then $\partial\text{ReLU}(\mathbf{x})$ denotes the coordinate-wise evaluation of $\partial\text{ReLU}$. Going back to monDEQ, let $\mathbf{s}$ be any subgradient of the implicit variable: $\mathbf{s} \in \partial\text{ReLU}(\mathbf{Wz} + \mathbf{Ux} + \mathbf{u})$. We can write equivalently a system of polynomial inequalities:

$$\mathbf{s}(\mathbf{s} - 1) \leq 0, \mathbf{s}(\mathbf{Wz} + \mathbf{Ux} + \mathbf{u}) \geq 0, (\mathbf{s} - 1)(\mathbf{Wz} + \mathbf{Ux} + \mathbf{u}) \geq 0. \qquad (5)$$

For the following discussion, $\partial\text{ReLU}$ will refer to the equivalent polynomial systems (5).

With the semialgebraicity of ReLU in (1) and $\partial\text{ReLU}$ in (5), one is able to compute upper bounds of the Lipschitz constant of $F$ via POPs. The proof of the next Lemma is postponed to Appendix A.1.

**Lemma 1** *Define*

$$\tilde{L}_{F,\mathcal{S}}^q = \max\{\mathbf{t}^T \mathbf{U}^T \mathbf{y} : \mathbf{t}, \mathbf{x} \in \mathbb{R}^{p_0}, \mathbf{s}, \mathbf{z}, \mathbf{y}, \mathbf{r} \in \mathbb{R}^p, \mathbf{v}, \mathbf{w} \in \mathbb{R}^K, \mathbf{x} \in \mathcal{S},$$

$$\|\mathbf{t}\|_q \leq 1, \mathbf{w}^T \mathbf{v} \leq 1, \|\mathbf{w}\|_q \leq 1, \mathbf{r} - \mathbf{W}^T \mathbf{y} = \mathbf{C}^T \mathbf{v}, \mathbf{y} = \text{diag}(\mathbf{s}) \cdot \mathbf{r};$$

$$\mathbf{s} \in \partial\text{ReLU}(\mathbf{Wz} + \mathbf{Ux} + \mathbf{u}), \mathbf{z} = \text{ReLU}(\mathbf{Wz} + \mathbf{Ux} + \mathbf{u})\}. \qquad \text{(LipMON)}$$

*Then $\tilde{L}_{F,\mathcal{S}}^q$ is an upper bound of the Lipschitz constant of $F$ w.r.t. the $L_q$ norm, i.e., $L_{F,\mathcal{S}}^q \leq \tilde{L}_{F,\mathcal{S}}^q$.*

Since problem (LipMON) is a POP, we also consider Shor's relaxation and denote by $\hat{L}_{F,\mathcal{S}}^q$ the upper bound of $\tilde{L}_{F,\mathcal{S}}^q$ by Shor's relaxation, i.e., $\tilde{L}_{F,\mathcal{S}}^q \leq \hat{L}_{F,\mathcal{S}}^q$. Define $\delta := \varepsilon \hat{L}_{F,\mathcal{S}}^q$. By equations (3), (4), if

$\mathcal{E} \subset \mathcal{S}$, using Lemma 1 and the fact that $\|\cdot\|_\infty \leq \|\cdot\|_q$, we have $\|F(\mathbf{x}) - F(\mathbf{x}_0)\|_\infty \leq \delta$, yielding the following criterion:

**Certification criterion**: Let $y_0$ be the label of $\mathbf{x}_0$. Define $\tau := F(\mathbf{x}_0)_{y_0} - \max_{k \neq y_0} F(\mathbf{x}_0)_k$. If $2\delta < \tau$, then the network $F$ is $\varepsilon$-robust at $\mathbf{x}_0$.

*Remark*: In order to avoid some possible numerical issues, we add some bound constraints and redundant constraints to problem (LipMON), see Appendix A.2 for details. Shor's relaxation of Lipschitz Model for DNNs has already been extensively investigated in [14, 8]. If one want to certify robustness for several input test examples, then one may choose $S$ to be a big ball containing all such examples with an additional margin of $\epsilon$. Choosing a big ball for all input points requires to solve only one optimization problem, while choosing one ball for each input point (say among $N$) requires to solve $N$ optimization problems, which is much more costly. In this case it is more favorable to use the Certification Model directly.

### 3.3 Ellipsoid Model

As above, the input region $\mathcal{E}$ is a neighborhood of input $\mathbf{x}_0 \in \mathbb{R}^{p_0}$ with radius $\varepsilon$ for the $L_q$ norm, i.e., $\mathcal{E} = \{\mathbf{x} \in \mathbb{R}^p : \|\mathbf{x} - \mathbf{x}_0\|_q \leq \varepsilon\}$. More generally, $\mathcal{E}$ could be other general semialgebraic sets, such as polytopes or zonotopes. Denote by $F(\mathcal{E})$ the image of $\mathcal{E}$ by $F$. In this section, we aim at finding a semialgebraic set $\mathcal{C}$ such that $F(\mathcal{E}) \subseteq \mathcal{C}$. We choose $\mathcal{C}$ to be an ellipsoid which can in turn be used for robustness certification. Our goal is to find such outer-approximation ellipsoid with minimum volume. Let $\mathcal{C} := \{\xi \in \mathbb{R}^K : \|\mathbf{Q}\xi + \mathbf{b}\|_2 \leq 1\}$ be an ellipsoid in the output space $\mathbb{R}^K$ parametrized by $\mathbf{Q} \in \mathbb{S}^K$ and $\mathbf{b} \in \mathbb{R}^K$, where $\mathbb{S}^K$ is the set of PSD matrices of size $K \times K$. The problem of finding the minimum-volume ellipsoid containing the image $F(\mathcal{E})$ can be formulated as

$$\max_{\mathbf{Q} \in \mathbb{S}^K, \mathbf{b} \in \mathbb{R}^K} \{\det(\mathbf{Q}) : \mathbf{x} \in \mathcal{E}, \|\mathbf{Q}(\mathbf{Cz} + \mathbf{c}) + \mathbf{b}\|_2 \leq 1, \mathbf{z} = \text{ReLU}(\mathbf{Wz} + \mathbf{Ux} + \mathbf{u})\}.$$

(EllipMON-POP)

By semialgebraicity of both ReLU in (1) and $\mathcal{C}$, problem (EllipMON-POP) can be cast as a POP (the determinant being a polynomial). We no longer apply Shor's relaxation, we rather replace the non-negativity output constraint $1 - \|\mathbf{Q}(\mathbf{Cz} + \mathbf{c}) + \mathbf{b}\|_2 \geq 0$ by a stronger Putinar's certificate related to both ReLU and input constraints. We can then relax the non-convex problem (EllipMON-POP) to a problem with SOS constraints, which can be reformulated by (convex) SDP constraints. This is due to the fact that a polynomial $f$ of degree at most $2d$ is SOS if and only if there exists a PSD matrix $\mathbf{M}$ (called a *Gram matrix*) such that $f(\mathbf{x}) = \mathbf{v}(\mathbf{x})^T \mathbf{M} \mathbf{v}(\mathbf{x})$, for all $\mathbf{x} \in \mathbb{R}^p$, with $\mathbf{v}(\mathbf{x})$ being the vector containing all monomials of degree at most $d$. In summary, we relax problem (EllipMON-POP) to an SOS constrained problem keeping the determinant unchanged:

$$\max_{\mathbf{Q} \in \mathbb{S}^K, \mathbf{b} \in \mathbb{R}^K} \{\det(\mathbf{Q}) : 1 - \|\mathbf{Q}(\mathbf{Cz} + \mathbf{c}) + \mathbf{b}\|_2^2 = \sigma_0(\mathbf{x}, \mathbf{z}) + \sigma_1(\mathbf{x}, \mathbf{z})^T g_q(\mathbf{x} - \mathbf{x}_0)$$
$$+ \tau(\mathbf{x}, \mathbf{z})^T (\mathbf{z}(\mathbf{z} - \mathbf{Wz} - \mathbf{Ux} - \mathbf{u})) + \sigma_2(\mathbf{x}, \mathbf{z})^T (\mathbf{z} - \mathbf{Wz} - \mathbf{Ux} - \mathbf{u}) + \sigma_3(\mathbf{x}, \mathbf{z})^T \mathbf{z}\}.$$

(EllipMON-SOS-$d$)

where $g_q(\mathbf{x}) = \varepsilon^q - \|\mathbf{x}\|_q^q$ for $q < +\infty$ and $g_q(\mathbf{x}) = \varepsilon^2 - |\mathbf{x}|^2$ for $q = +\infty$, $\sigma_0$ is a vector of SOS polynomials of degree at most $2d$, $\sigma_1$ is a vector of SOS polynomials of degree at most $2(d - \lceil q/2 \rceil)$ for $q < +\infty$ and $2d - 2$ for $q = +\infty$, $\sigma_2, \sigma_3$ are vectors of SOS polynomials of degrees at most $2d - 2$, $\tau$ is a vector of polynomials of degree at most $2d - 2$. Problem (EllipMON-SOS-$d$) provides an ellipsoid feasible for (EllipMON-POP), that is an ellipsoid which contains $F(\mathcal{E})$. In practice, the determinant is replaced by a log-det objective because there exist efficient solver dedicated to optimize such objectives on SDP constraints. By increasing the relaxation order $d$ in problem (EllipMON-SOS-$d$), one is able to obtain a hierarchy of log-det objected SDP problems for which the outer-approximation ellipsoids have decreasing volumes.

In this paper, we only consider the case $p = 2, \infty$, and order-1 relaxation ($d = 1$). Therefore, $\sigma_i$ and $\tau$ are all (vectors of) real (non-negative) numbers for $i = 1, 2, 3$, except that $\sigma_0$ is an SOS polynomial of degree at most 2. In this case, problem (EllipMON-SOS-$d$) is equivalent to a problem with log-det objective and SDP constraints, as the following lemma states (proof postponed to Appendix A.3):

**Lemma 2** *For $p = 2$ or $p = \infty$, problem* (EllipMON-SOS-$d$) *with $d = 1$ is equivalent to*

$$\max_{\mathbf{Q} \in \mathbb{S}^K, \mathbf{b} \in \mathbb{R}^K, \sigma_1, \sigma_2, \sigma_3 \geq 0, \tau \in \mathbb{R}^p} \{\log \det(\mathbf{Q}) : -\mathbf{M} \succeq 0\}.$$

(EllipMON-SDP)

*where* $\mathbf{M} \in \mathbb{S}^{(p_0+p+1)\times(p_0+p+1)}$ *is a symmetric matrix parametrized by the decision variables* $(\mathbf{Q}, \mathbf{b})$, *the coefficients* $(\sigma_1, \sigma_2, \sigma_3, \tau)$, *and the parameters of the network* $(\mathbf{W}, \mathbf{U}, \mathbf{u}, \mathbf{C}, \mathbf{c})$.

Since the outer-approximation ellipsoid $\mathcal{C} = \{\xi \in \mathbb{R}^K : \|\mathbf{Q}\xi + \mathbf{b}\|_2 \leq 1\}$ contains the image $F(\mathcal{E})$, i.e., all possible outputs of the input region $\mathcal{E}$, one is able to certify robustness by solving the following optimization problems.

**Certification criterion**: Let $y_0$ be the label of $\mathbf{x}_0$. For $i \neq y_0$, define $\delta_i := \max_{\xi \in \mathbb{R}^K}\{\xi_i - \xi_{y_0} : \|\mathbf{Q}\xi + \mathbf{b}\|_2 \leq 1\}$. If $\delta_i < 0$ for all $i \neq y_0$, then the network $F$ is $\varepsilon$-robust at $\mathbf{x}_0$.

The certification criterion for Ellipsoid Model has a geometric explanation: for $i \neq y_0$, denote by $\mathcal{P}_i$ the projection map from output space $\mathbb{R}^K$ to its 2-dimensional subspace $\mathbb{R}_{y0} \times \mathbb{R}_i$, i.e., $\mathcal{P}_i(\xi) = [\xi_{y_0}, \xi_i]^T$ for all $\xi \in \mathbb{R}^K$. Let $\mathcal{L}_i$ be the line in subspace $\mathbb{R}_{y0} \times \mathbb{R}_i$ defined by $\{[\xi_{y_0}, \xi_i]^T \in \mathbb{R}_{y0} \times \mathbb{R}_i : \xi_{y_0} = \xi_i\}$. Then the network $F$ is $\varepsilon$-robust if the projection $\mathcal{P}_i(\mathcal{C})$ lies strictly below the line $\mathcal{L}_i$ for all $i \neq y_0$. We give an explicit example in Section 4.3 to visually illustrate this.

### 3.4  Summary of the Models

We have already presented three models which can all be dedicated to certify robustness of neural networks. However, the size and complexity of each model are different. We summarize the number of variables in each model and the maximum size of PSD matrices in the resulting Shor's relaxation, see Table 1. The complexity of our models only depends on the number of neurons in the input layer and implicit layer. The size of PSD matrices is a limiting factor for SDP solvers, our models are practically restricted to network for which such size can be handled by SDP solvers. For the popular dataset MNIST [46], whose input dimension is $28 \times 28 = 784$, we are able to apply our model on monDEQs with moderate size implicit layers (87) and report the corresponding computation time.

Table 1: Summary of the number of variables the three models

|  | Robustness Model | Lipschitz Model | Ellipsoid Model |
|---|---|---|---|
| Num. of variables | $p_0 + p$ | $2p_0 + 4p + 2K$ | $p_0 + 3p + K + K^2$ |
| Max. size of PSD matrices | $1 + p_0 + p$ | $1 + 2p_0 + 4p + 2K$ | $1 + p_0 + p$ |

## 4  Experiments

In this section, we present the experimental results of Robustness Model, Lipschitz Model and Ellipsoid Model described in Section 3 for a pretrained monDEQ on MNIST dataset. The network we use consists of a fully-connected implicit layer with 87 neurons and we set its monotonicity parameter $m$ to be 20. The training hyperparameters are set to be the same as in Table D1 of [45], where the training code (in Python) is available at `https://github.com/locuslab/monotone_op_net`. Training is based on the normalized MNIST database in [45], we use the same normalization setting on each test example with mean $\mu = 0.1307$ and standard deviation $\sigma = 0.3081$, which means that each input is an image of size $28 \times 28$ with entries varying from $-0.42$ to $2.82$. And for every perturbation $\varepsilon$, we also take the normalization into account, i.e., we use the normalized perturbation $\varepsilon/\sigma$ for each input.

Since all our three models can be applied to certify robustness of neural networks, we first compare the performance of each model in certification of the first 100 test MNIST examples. Then we compare the upper bounds of Lipschitz Model with the upper bounds proposed in [33]. Finally we show that Ellipsoid Model can also be applied for reachability analysis. For Certification model and Lipschitz model, we implement them in Julia [5] with JuMP [11] package; for Ellipsoid model, we implement it in Matlab [39] with CVX [17] package. For all the three models, we use Mosek [29] as a backend to solve the targeted POPs. All experiments are performed on a personal laptop with an Intel 8-Core i7-8665U CPU @ 1.90GHz Ubuntu 18.04.5 LTS, 32GB RAM. The code of all our models is available at `https://github.com/NeurIPS2021Paper4075/SemiMonDEQ`.

## 4.1 Robustness certification

We consider $\varepsilon = 0.1$ for the $L_2$ norm and $\varepsilon = 0.1, 0.05, 0.01$ for the $L_\infty$ norm. For each model, we compute the ratio of certified test examples among the first 100 test inputs. Following [33], we also compute the *projected gradient descent (PGD)* attack accuracy using Foolbox library [37], which indicates the ratio of non-successful attacks among our 100 inputs. Note that the ratio of certified examples should always be less or equal than the ratio of non-successful attacks. The gaps between them shows how many test examples there are for which we are neither able to certify robustness nor find adversarial attacks.

*Remark*: For Lipschitz Model, we use inequality (3) to test robustness, i.e., we compute directly the upper bound of the Lipschitz constant of $F$ rather than $\mathbf{z}$ (seen as a function of $\mathbf{x}$) where $\mathcal{S}$ is a big ball containing all test examples.

From Table 2, we see that the monDEQ is robust to all the 100 test examples for the $L_2$ norm and $\varepsilon = 0.1$ (the only example that we can not certify is because the label itself is wrong). However, it is not robust for the $L_\infty$ norm at the same level of perturbation (all our three models cannot certify any examples), and the PGD algorithm finds adversarial examples for 85% of the inputs. The network becomes robust again for the $L_\infty$ norm when we reduce the perturbation $\varepsilon$ to 0.01. Overall, we see that Robustness Model is the best model as it provides the highest ratio, Ellipsoid Model is the second best model compared to Robustness Model, and Lipschitz Model provides the lowest ratio. As a trade-off, for each test example, Robustness Model requires to consider at most 9 optimization problems, each one being solved in around 150 seconds, while Ellipsoid Model requires to consider only one problem, which is solved in around 500 seconds. We only need to calculate one (global) Lipschitz constant, which takes around 1500 seconds, so that we are able to certify any number of inputs. Each model we propose provide better or equal certification accuracy compared to [33], and significant improvements for $L^\infty$ perturbations.

Table 2: Ratio of certified test examples and running time per example by different methods. We consider $L_2$ norm with $\varepsilon = 0.1$ and $L_\infty$ norm with $\varepsilon = 0.1, 0.05, 0.01$. The ratio is based on the first 100 MNIST test examples, and we count the average computation time (with unit second) for one example of each method. The ratio in parentheses of the column "Lipschitz Model" are computed by the Lipschitz constant given in [33] (see Section 4.2 for details). Exact binomial 95% confidence intervals are given in bracket.

| Norm | $\varepsilon$ | Robustness Model (1350s / example) | Lipschitz Model (1500s in total) | Ellipsoid Model (500s / example) | PGD Attack |
|---|---|---|---|---|---|
| $L_2$ | 0.1 | 99% [>94] | 91% (91% ) [>83] | 99% [>94] | 99% [>94] |
| $L_\infty$ | 0.1 | 0% [<4] | 0% (0%) [<4] | 0% [<4] | 15% [8, 24] |
| | 0.05 | 24% [16, 34] | 0% (0%) [<4] | 0% [<4] | 82% [73, 89] |
| | 0.01 | 99% [>94] | 24% [16, 34] (0%) [<4] | 92% [>84] | 99% [>94] |

Figure 2 in Appendix A.4 shows the original image of the first test example (2a) and an adversarial attack (2b) for the $L_\infty$ norm with $\varepsilon = 0.1$ found by the PGD algorithm in [37].

## 4.2 Comparison with Lipschitz constants

In this section, we compare the upper bounds of Lipschitz constants computed by Lipschitz Model with the upper bounds proposed in [33]. Notice that the upper bounds in [33] only involve the function $\mathbf{z}(\mathbf{x})$, hence we are only able to use inequality (4) to test robustness. In fact the quantity $\|\!|\mathbf{C}|\!\|_q \cdot L_{\mathbf{z},\mathcal{S}}^q$ can be regarded as an upper bound of $L_{F,\mathcal{S}}^q$, the Lipschitz constant of $F$. We denote by $\mathbf{UB}_{\mathbf{z}}^2$ the upper bound of the Lipschitz constant of $\mathbf{z}$ w.r.t. the $L_2$ norm, given by $\mathbf{UB}_{\mathbf{z}}^2 = \|\!|\mathbf{U}|\!\|_2/m$ according to [33], where $\mathbf{U}$ is the parameter of the network and $m$ is the monotonicity factor. We can then compute the upper bound w.r.t. the $L_\infty$ norm by $\mathbf{UB}_z^\infty = \sqrt{p_0} \cdot \mathbf{UB}_z^2$ where $p_0$ is the input dimension. The upper bound of Lipschitz constant of $F$ is computed via the upper bound of $\mathbf{z}$: $\mathbf{UB}_F^q = \|\!|\mathbf{C}|\!\|_q \cdot \mathbf{UB}_{\mathbf{z}}^q$. Denote similarly by $\mathbf{SemiUB}_F^q$ the upper bounds of Lipschitz constants of $F$ provided by Lipschitz Model, w.r.t. the $L_q$ norm.

Table 3: Comparison of upper bounds of Lipschitz constant for $L_2$ and $L_\infty$ norm, and the corresponding computation time (with unit second).

| | $q = 2$ | | $q = \infty$ | |
|---|---|---|---|---|
| | bound | time (s) | bound | time (s) |
| $\mathbf{UB}_F^q$ | 4.80 | - | 824.14 | - |
| $\mathbf{SemiUB}_F^q$ | 4.67 | 1756.58 | 108.84 | 1898.65 |

From Table 3, we see that Lipschitz Model provides consistently tighter upper bounds than the ones in [33]. Especially for $L_\infty$ norm, the upper bound computed by $\|\|\mathbf{C}\|\|_\infty \cdot \mathbf{UB}_\mathbf{z}^\infty$ is rather crude compared to the bound obtained directly by Lipschitz Model. Therefore, we are able to certify more examples using $\mathbf{SemiUB}_F^q$ than $\mathbf{UB}_F^q$, see Table 2.

### 4.3 Outer ellipsoid approximation

In this section, we provide a visible illustration of how Ellipsoid Model can be applied to certify robustness of neural networks.

Take the first MNIST test example (which is classified as 7) for illustration. For $\varepsilon = 0.1$, this example is certified to be robust for the $L_2$ norm but not for the $L_\infty$ norm. We show the landscape of the projections onto $\mathbb{R}_7 \times \mathbb{R}_3$, i.e., the $x$-axis indicates label 7 and the $y$-axis indicates label 3. In Figure 1, the red points are projections of points in the image $F(\mathcal{E})$, for $\mathcal{E}$ an $L_2$ or $L_\infty$ norm perturbation zone, the black circles are projections of some (successful and unsuccessful) adversarial examples found by the PGD algorithm. Notice that the adversarial examples also lie in the image $F(\mathcal{E})$. The blue curve is the boundary of the projection of the outer-approximation ellipsoid (which is an ellipse), and the blue dashed line plays the role of a certification threshold. Figure 1a shows the landscape for the $L_2$ norm, we see that the ellipse lies strictly below the threshold line, which means that for all points $\xi \in \mathcal{C}$, we have $\xi_3 < \xi_7$. Hence for all $\xi \in F(\mathcal{E})$, we also have $\xi_3 < \xi_7$. On the other hand, for the $L_\infty$ norm, we see from Figure 1b that the threshold line crosses the ellipse, which means that we are not able to certify robustness of this example by Ellipsoid Model. Indeed, we can find adversarial examples with the PGD algorithm, as shown in Figure 1b by the black circles that lie above the threshold line. The visualization of one of the attack examples is shown in Figure 2 in Appendix A.4.

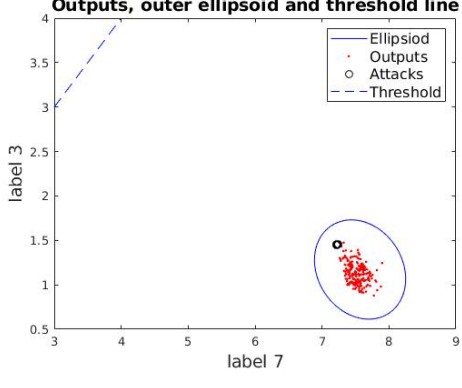
(a) Certified example for the $L_2$ norm

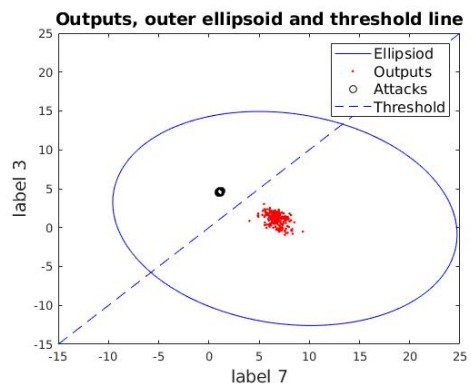
(b) Non-certified example for the $L_\infty$ norm

Figure 1: Visualization of the outer-approximation ellipsoids and outputs with $\varepsilon = 0.1$ for $L_2$ norm (left) and $L_\infty$ norm (right). The red points are image of the input region, the blue curve is the ellipsoid we compute, the blue dashed line is the threshold line used for certifying robustness of inputs, and the black circles are attack examples found by PGD algorithm.

# 5 Conclusion and Future Works

In this paper, we introduce semialgebraic representations of monDEQ and propose several POP models that are useful for certifying robustness, estimating Lipschitz constants and computing outer-approximation ellipsoids. For each model, there are several hierarchies of relaxations that allow us to improve the results by increasing the relaxation order. Even though we simply consider the order-1 relaxation, we obtain tighter upper bounds of Lipschitz constants compared to the results in [33]. Consequently, we are able to certify robustness of more examples.

Our models are based on SDP relaxation, hence requires an efficient SDP solver. However, the stat-of-the-art SDP solver Mosek (by interior-point method) can only handle PSD matrices of moderate size (smaller than 5000). This is the main limitation of our method if the dimension of the input gets larger. Moreover, we only consider the fully-connected monDEQ based on MNIST datasets for illustration. One important and interesting future work is to generalize our model to single and multi convolutional monDEQ, and to other datasets such as CIFAR [26] and SVHN [32]. Directly using off-the-shelves interior point SDP solvers to solve the problems for convolutional networks is not possible because of their current size limitation. Fortunately, a convolutional layer can be regarded as a fully-connected layer with a larger (but sparse) weight matrix. Hence one is able to build similar models via sparse polynomial optimization tools.

The authors in [10] provide an interesting and promising first-order method as an alternative to the costly interior point methods for solving semidefinite relaxations associated with robustness certification of DNNs. What is crucial in [10] is to exploit the network layer structure for efficient back-propagation in gradient computation. In particular, the algorithm requires memory only linear in the total number of network activations and only requires a fixed number of forward/backward passes through the network per iteration. This enables the algorithm to certify robustness of large-scaled networks more efficiently and accurately. However, the auto-differentiation technique for computing subgradients in DNNs does not apply directly to monDEQs, since computing subgradients in monDEQs involves solving fixed-point equations. Therefore, adapting the approach of [10] for solving more efficiently our semidefinite relaxations associated with monDEQs is certainly worth considering but not straightforward. It is a topic of further investigation.

The authors in [33] showed that we can train DEQs with small Lipschitz constants for the $L_2$ norm, by controlling the monotonicity of the weight matrix. This guarantees the robustness of monDEQ w.r.t. the $L_2$ norm but not for the $L_\infty$ norm. A natural investigation track is to adapt this training technique to the $L_\infty$ norm with a better control of the associated Lipschitz constant.

## Acknowledgments and Disclosure of Funding

The authors acknowledge the support of AI Interdisciplinary Institute ANITI funding, through the French "Investing for the Future – PIA3" program under the Grant agreement ANR-19-PI3A-0004. Edouard Pauwels acknowledges the support of Air Force Office of Scientific Research, Air Force Material Command, USAF, under grant numbersFA9550-19-1-702 6and ANR MaSDOL 19-CE23-0017-01. This work was supported by the National Research Foundation, Prime Minister's Office, Singapore under its Campus for Research Excellence and Technological Enterprise (CREATE) programme. Victor Magron was supported by the FMJH Program PGMO (EPICS project) and EDF, Thales, Orange et Criteo, as well as from the Tremplin ERC Stg Grant ANR-18-ERC2-0004-01 (T-COPS project). This work has benefited from the European Union's Horizon 2020 research and innovation programme under the Marie Sklodowska-Curie Actions.

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
