# A   Appendix

This is the appendix for "Semialgebraic Representation of Monotone Deep Equilibrium Models and Applications to Certification".

## A.1   Proof of Lemma 1

**Definition 1** *(Clarke's generalized Jacobian)* [10] *Let $f : \mathbb{R}^n \to \mathbb{R}^m$ be a locally Lipschitz vector-valued function, denote by $\Omega_f$ any zero measure set such that $f$ is differentiable outside $\Omega_f$. For $\mathbf{x} \notin \Omega_f$, denote by $\mathcal{J}_f(\mathbf{x})$ the Jacobian matrix of $f$ evaluated at $\mathbf{x}$. For any $\mathbf{x} \in \mathbb{R}^n$, the generalized Jacobian, or Clarke Jacobian, of $f$ evaluated at $\mathbf{x}$, denoted by $\mathcal{J}_f^C(\mathbf{x})$, is defined as the convex hull of all $m \times n$ matrices obtained as the limit of a sequence of the form $\mathcal{J}_f(\mathbf{x}_i)$ with $\mathbf{x}_i \to \mathbf{x}$ and $\mathbf{x}_i \notin \Omega_f$. Symbolically, one has*

$$\mathcal{J}_f^C(\mathbf{x}) := \operatorname{conv}\{\lim \mathcal{J}_f(\mathbf{x}_i) : \mathbf{x}_i \to \mathbf{x}, \ \mathbf{x}_i \notin \Omega_f\}.$$

In order to estimate the Lipschitz constant $L_{F,\mathcal{S}}^q$, we need the following lemma:

**Lemma 3** *Let $F : \mathbb{R}^{p_0} \to \mathbb{R}^K, \mathbf{x} \mapsto \mathbf{C}\mathbf{z}(\mathbf{x})$ be the fully-connected monDEQ. Its Lipschitz constant is upper bounded by the supremum of the operator norm of its generalized Jacobian, i.e., define*

$$\bar{L}_{F,\mathcal{S}}^q := \sup_{\mathbf{t},\mathbf{x} \in \mathbb{R}^{p_0}, \mathbf{v},\mathbf{w} \in \mathbb{R}^K, \mathbf{J} \in \mathcal{J}_\mathbf{z}^C(\mathbf{x})} \{\mathbf{t}^T \mathbf{J}^T \mathbf{C}^T \mathbf{v} : \|\mathbf{t}\|_q \le 1, \ \mathbf{w}^T \mathbf{v} \le 1, \ \|\mathbf{w}\|_q \le 1, \ \mathbf{x} \in \mathcal{S}\}, \quad (6)$$

*then $L_{F,\mathcal{S}}^q \le \bar{L}_{F,\mathcal{S}}^q$.*

**Proof :** Since $\mathbf{z}(\mathbf{x}) = \mathrm{ReLU}(\mathbf{W}\mathbf{z}(\mathbf{x}) + \mathbf{U}\mathbf{x} + \mathbf{u})$ by definition of monDEQ, $\mathbf{z}(\mathbf{x})$ is Lipschitz according to [34, Theorem 1]. Furthermore, $\mathbf{z}(\mathbf{x})$ is semialgebraic by the semialgebraicity of $\mathrm{ReLU}$ in (1). Therefore, the Clarke Jacobian of $\mathbf{z}$ is conservative. Indeed by [10, Proposition 2.6.2], the Clarke Jacobian is included in the product of subgradients of its coordinates which is a conservative field by [7, Lemma 3, Theorems 2 and 3]. Since $F = \mathbf{C} \circ \mathbf{z}$, the mapping $\mathbf{C}\mathcal{J}_\mathbf{z}^C : \mathbf{x} \rightrightarrows \mathbf{C}\mathbf{J}$, where $\mathbf{J} \in \mathcal{J}_\mathbf{z}^C$, is conservative for $F$ by [7, Lemma 5]. So it satisfies an integration formula along segments. Let $\mathbf{x}_1, \mathbf{x}_2 \in \mathcal{E}$, and let $\gamma : [0,1] \to \mathbb{R}^{p_0}$ be a parametrization of the segment defined by $\gamma(t) = \mathbf{x}_1 + t(\mathbf{x}_2 - \mathbf{x}_1)$ (which is absolutely continuous). For almost all $t \in [0,1]$, we have $\frac{\mathrm{d}}{\mathrm{d}t}F(\gamma(t)) = \mathbf{C}\mathbf{J}\gamma'(t) = \mathbf{C}\mathbf{J}(\mathbf{x}_2 - \mathbf{x}_1)$ for all $\mathbf{J} \in \mathcal{J}_\mathbf{z}^C(\gamma(t))$.

Let $M = \sup_{\mathbf{x} \in \mathcal{S}, \mathbf{J} \in \mathcal{J}_\mathbf{z}^C(\mathbf{x})} \|\|\mathbf{C}\mathbf{J}\|\|_q$ be the supremum of the operator norm $\|\|\mathbf{C}\mathbf{J}\|\|_q$ for all $\mathbf{J} \in \mathcal{J}_\mathbf{z}^C(\mathbf{x})$ and all $\mathbf{x} \in \mathcal{S}$. We prove that $M < +\infty$. Indeed, $\mathbf{z}(\mathbf{x})$ is Lipschitz, hence there exists $N > 0$ such that $\|\|\mathbf{J}\|\|_q < N$ for all $\mathbf{J} \in \mathcal{J}_\mathbf{z}^C(\mathbf{x})$ and all $\mathbf{x} \in \mathcal{S}$. The value $M$ is thus upper bounded by $\|\|\mathbf{C}\|\|_q N$.

Therefore, for almost all $t \in [0,1]$, $\|\frac{\mathrm{d}}{\mathrm{d}t}F(\gamma(t))\|_q \le M\|\mathbf{x}_2 - \mathbf{x}_1\|_q$, and by integration,

$$\|F(\mathbf{x}_2) - F(\mathbf{x}_1)\|_q = \left\| \int_0^1 \frac{\mathrm{d}}{\mathrm{d}t}F(\gamma(t))\mathrm{d}t \right\|_q \le \int_0^1 \left\| \frac{\mathrm{d}}{\mathrm{d}t}F(\gamma(t)) \right\|_q \mathrm{d}t \le M\|\mathbf{x}_2 - \mathbf{x}_1\|_q, \quad (7)$$

which proves that $L_{F,\mathcal{S}}^q \le M$. Let us show that $M = \bar{L}_{F,\mathcal{S}}^q$. Fix $\mathbf{x} \in \mathbb{R}^{p_0}$ and $\mathbf{J} \in \mathcal{J}_\mathbf{z}^C(\mathbf{x})$. By the definition of operator norm,

$$\|\|\mathbf{C}\mathbf{J}\|\|_q = \|\|(\mathbf{C}\mathbf{J})^T\|\|_q^* = \max_{\mathbf{v} \in \mathbb{R}^K}\{\|\mathbf{J}^T\mathbf{C}^T\mathbf{v}\|_q^* : \|\mathbf{v}\|_q^* \le 1\}$$

$$= \max_{\mathbf{t} \in \mathbb{R}^{p_0}, \mathbf{v} \in \mathbb{R}^K}\{\mathbf{t}^T\mathbf{J}^T\mathbf{C}^T\mathbf{v} : \|\mathbf{t}\|_q \le 1, \ \|\mathbf{v}\|_q^* \le 1\}$$

$$= \max_{\mathbf{t} \in \mathbb{R}^{p_0}, \mathbf{v},\mathbf{w} \in \mathbb{R}^K}\{\mathbf{t}^T\mathbf{J}^T\mathbf{C}^T\mathbf{v} : \|\mathbf{t}\|_q \le 1, \ \mathbf{w}^T\mathbf{v} \le 1, \ \|\mathbf{w}\|_q \le 1\}, \quad (8)$$

where $\| \cdot \|_q^*$ denotes the dual norm of $\| \cdot \|_q$ defined by $\|\mathbf{v}\|_q^* := \sup_{\mathbf{w} \in \mathbb{R}^K}\{\mathbf{w}^T\mathbf{v} : \|\mathbf{w}\|_q \le 1\}$ for all $\mathbf{v} \in \mathbb{R}^K$, and the first equality is due to the fact that the operator norm of matrix $\mathbf{C}\mathbf{J}$ induced by norm $\| \cdot \|_q$ is equal to the operator norm of its transpose $(\mathbf{C}\mathbf{J})^T$ induced by the dual norm $\| \cdot \|_q^*$.

Indeed, by definition of operator norm and dual norm, we have

$$\|\mathbf{CJ}\|_q = \sup_{\mathbf{x}\in\mathbb{R}^{p_0}} \{\|\mathbf{CJx}\|_q : \|\mathbf{x}\|_q \leq 1\} = \sup_{\mathbf{x}\in\mathbb{R}^{p_0},\mathbf{y}\in\mathbb{R}^p} \{\mathbf{y}^T\mathbf{CJx} : \|\mathbf{x}\|_q \leq 1, \|\mathbf{y}\|_q^* \leq 1\}$$

$$= \sup_{\mathbf{x}\in\mathbb{R}^{p_0},\mathbf{y}\in\mathbb{R}^p} \{\mathbf{x}^T(\mathbf{CJ})^T\mathbf{y} : \|\mathbf{x}\|_q \leq 1, \|\mathbf{y}\|_q^* \leq 1\} = \sup_{\mathbf{y}\in\mathbb{R}^p} \{\|(\mathbf{CJ})^T\mathbf{y}\|_q^* : \|\mathbf{y}\|_q^* \leq 1\}$$

$$= \||(\mathbf{CJ})^T\||_q^* .$$

The quantity $\bar{L}_{F,\mathcal{S}}^q$ is just the maximization of Equation (8) for all $\mathbf{x} \in \mathbb{R}^{p_0}$ and all $\mathbf{J} \in \mathcal{J}_{\mathbf{z}}^C(\mathbf{x})$ and therefore equals $M$. $\qquad\square$

The function $\mathbf{z}$ is semialgebraic, and therefore, there exists a closed zero measure set $\Omega_{\mathbf{z}}$ such that $\mathbf{z}$ is continuously differentiable on the complement of $\Omega_{\mathbf{z}}$. For any $\mathbf{x} \notin \Omega_{\mathbf{z}}$, since $\mathbf{z}$ is $C^1$ at $\mathbf{x}$, we have $\mathcal{J}_{\mathbf{z}}^C(\mathbf{x}) = \{\mathcal{J}_{\mathbf{z}}(\mathbf{x})\}$ by definition of the Clarke Jacobian. Fix $\mathbf{x} \notin \Omega_{\mathbf{z}}$ arbitrary. According to the Corollary of Theorem 2.6.6, on page 75 of [10], we have

$$\mathcal{J}_{\mathbf{z}}^C(\mathbf{x}) \subseteq \mathrm{conv}\{\mathcal{J}_{\mathrm{ReLU}}^C(\mathbf{Wz}(\mathbf{x}) + \mathbf{Ux} + \mathbf{u}) \cdot \mathcal{J}_{\mathbf{Wz}(\mathbf{x})+\mathbf{Ux}+\mathbf{u}}^C(\mathbf{x})\}$$

$$= \mathrm{conv}\{\mathcal{J}_{\mathrm{ReLU}}^C(\mathbf{Wz}(\mathbf{x}) + \mathbf{Ux} + \mathbf{u}) \cdot (\mathbf{W} \cdot \mathcal{J}_{\mathbf{z}}(\mathbf{x}) + \mathbf{U})\}$$

$$= \mathcal{J}_{\mathrm{ReLU}}^C(\mathbf{Wz}(\mathbf{x}) + \mathbf{Ux} + \mathbf{u}) \cdot (\mathbf{W} \cdot \mathcal{J}_{\mathbf{z}}(\mathbf{x}) + \mathbf{U}), \tag{9}$$

where the first inclusion is from the cited Corollary, the first equality is because $\mathbf{z}$ is $C^1$ at $\mathbf{x}$ so that the chain rule applies, and the last one is because the Clarke Jacobian is convex.

Fix any any $\bar{\mathbf{x}} \in \mathbb{R}^{p_0}$, then by definition $\mathcal{J}_{\mathbf{z}}^C(\bar{\mathbf{x}}) = \mathrm{conv}\{\lim \mathcal{J}_{\mathbf{z}}(\mathbf{x}_i) : \mathbf{x}_i \to \bar{\mathbf{x}}, i \to +\infty, \mathbf{x}_i \notin \Omega_{\mathbf{z}}\}$. Let $\{\mathbf{x}_i\}_{i\in\mathbb{N}}$ be a sequence not in $\Omega_{\mathbf{z}}$ converging to $\bar{\mathbf{x}}$, for each $\mathbf{x}_i \notin \Omega_{\mathbf{z}}$, we have by (9) that $\mathcal{J}_{\mathbf{z}}(\mathbf{x}_i) \in \mathcal{J}_{\mathrm{ReLU}}^C(\mathbf{Wz}(\mathbf{x}_i) + \mathbf{Ux}_i + \mathbf{u}) \cdot (\mathbf{W} \cdot \mathcal{J}_{\mathbf{z}}(\mathbf{x}_i) + \mathbf{U})$, i.e., there exists $\mathbf{Y}_i \in \mathcal{J}_{\mathrm{ReLU}}^C(\mathbf{Wz}(\mathbf{x}_i) + \mathbf{Ux}_i + \mathbf{u})$ such that $\mathcal{J}_{\mathbf{z}}(\mathbf{x}_i) = \mathbf{Y}_i(\mathbf{W} \cdot \mathcal{J}_{\mathbf{z}}(\mathbf{x}_i) + \mathbf{U})$. By [10, proposition 2.6.2 (b)], $\mathcal{J}_{\mathrm{ReLU}}^C$ has closed graph. Therefore, by continuity of $\mathbf{z}$, up to a subsequence, $\mathbf{Y}_i \to \mathbf{Y} \in \mathcal{J}_{\mathrm{ReLU}}^C(\mathbf{Wz}(\bar{\mathbf{x}}) + \mathbf{U}\bar{\mathbf{x}} + \mathbf{u})$ for $i \to +\infty$, which means

$$\mathcal{J}_{\mathbf{z}}^C(\bar{\mathbf{x}}) \subseteq \{\mathbf{J} : \mathbf{Y} \in \mathcal{J}_{\mathrm{ReLU}}^C(\mathbf{Wz}(\bar{\mathbf{x}}) + \mathbf{U}\bar{\mathbf{x}} + \mathbf{u}), \mathbf{J} = \mathbf{Y}(\mathbf{WJ} + \mathbf{U})\}, \tag{10}$$

for all $\bar{\mathbf{x}} \in \mathbb{R}^{p_0}$. Let $\mathbf{Y} \in \mathcal{J}_{\mathrm{ReLU}}^C(\mathbf{Wz} + \mathbf{Ux} + \mathbf{u})$, since we have coordinate-wise applications of ReLU, we have that $\mathbf{Y} = \mathrm{diag}(\mathbf{s})$ with $\mathbf{s} \in \partial\mathrm{ReLU}(\mathbf{Wz} + \mathbf{Ux} + \mathbf{u})$. By equation (10), the right-hand side of equation (6) is upper bounded by

$$\max_{\mathbf{t},\mathbf{x}\in\mathbb{R}^{p_0},\mathbf{s},\mathbf{z}\in\mathbb{R}^p,\mathbf{v},\mathbf{w}\in\mathbb{R}^K,\mathbf{J}\in\mathbb{R}^{p\times p_0}} \{\mathbf{t}^T\mathbf{J}^T\mathbf{C}^T\mathbf{v} : \|\mathbf{t}\|_q \leq 1, \mathbf{w}^T\mathbf{v} \leq 1, \|\mathbf{w}\|_q \leq 1, \mathbf{x} \in \mathcal{S},$$

$$\mathbf{s} \in \partial\mathrm{ReLU}(\mathbf{Wz} + \mathbf{Ux} + \mathbf{u}), \mathbf{z} = \mathrm{ReLU}(\mathbf{Wz} + \mathbf{Ux} + \mathbf{u}),$$

$$\mathbf{J} = \mathrm{diag}(\mathbf{s}) \cdot (\mathbf{W} \cdot \mathbf{J} + \mathbf{U})\}. \tag{LipMON-a}$$

Notice that in problem (LipMON-a), we have a matrix variable $\mathbf{J}$ of size $p \times p_0$, i.e., containing $p \times p_0$ many variables, which is too large for any SDP solvers. To reduce the size, we use the *vector-matrix product* trick introduced in [46] to reduce the size of the unknown variables. From equation $\mathbf{J} = \mathrm{diag}(\mathbf{s}) \cdot (\mathbf{W} \cdot \mathbf{J} + \mathbf{U})$, we have $\mathbf{J} = (\mathbf{I}_p - \mathrm{diag}(\mathbf{s}) \cdot \mathbf{W})^{-1} \cdot \mathrm{diag}(\mathbf{s}) \cdot \mathbf{U}$. This inversion makes sense because of the strong monotonicity of $\mathbf{I}_p - \mathbf{W}$ and the fact that all entries of $\mathbf{s}$ lie in $[0, 1]$ [46, Proposition 1]. Hence

$$\mathbf{v}^T\mathbf{CJ} = \mathbf{v}^T\mathbf{C} \cdot (\mathbf{I}_p - \mathrm{diag}(\mathbf{s}) \cdot \mathbf{W})^{-1} \cdot \mathrm{diag}(\mathbf{s}) \cdot \mathbf{U} = \mathbf{r}^T \cdot \mathrm{diag}(\mathbf{s}) \cdot \mathbf{U}, \tag{11}$$

where $\mathbf{r}^T = \mathbf{v}^T\mathbf{C} \cdot (\mathbf{I}_p - \mathrm{diag}(\mathbf{s}) \cdot \mathbf{W})^{-1}$, which means $\mathbf{r} - \mathbf{W}^T \cdot \mathrm{diag}(\mathbf{s}) \cdot \mathbf{r} = \mathbf{C}^T\mathbf{v}$. Set $\mathbf{y} = \mathrm{diag}(\mathbf{s}) \cdot \mathbf{r}$ and transpose both sides of equation (11), we have $\mathbf{J}^T\mathbf{C}^T\mathbf{v} = \mathbf{U}^T\mathbf{y}$ with $\mathbf{r} - \mathbf{W}^T \cdot \mathbf{y} = \mathbf{C}^T\mathbf{v}$. We can then rewrite the objective function of (LipMON-a) as $\mathbf{t}^T\mathbf{U}^T\mathbf{y}$, leading to the following equivalent problem

$$\max_{\mathbf{t},\mathbf{x}\in\mathbb{R}^{p_0},\mathbf{s},\mathbf{z},\mathbf{y},\mathbf{r}\in\mathbb{R}^p,\mathbf{v},\mathbf{w}\in\mathbb{R}^K} \{\mathbf{t}^T\mathbf{U}^T\mathbf{y} : \|\mathbf{t}\|_q \leq 1, \mathbf{w}^T\mathbf{v} \leq 1, \|\mathbf{w}\|_q \leq 1, \mathbf{x} \in \mathcal{S},$$

$$\mathbf{s} \in \partial\mathrm{ReLU}(\mathbf{Wz} + \mathbf{Ux} + \mathbf{u}), \mathbf{z} = \mathrm{ReLU}(\mathbf{Wz} + \mathbf{Ux} + \mathbf{u}),$$

$$\mathbf{r} - \mathbf{W}^T\mathbf{y} = \mathbf{C}^T\mathbf{v}, \mathbf{y} = \mathrm{diag}(\mathbf{s}) \cdot \mathbf{r}\}. \tag{LipMON-b}$$

We have shown that (LipMON-b) is the right hand side of Equation (LipMON) in Lemma 1 and is an upper bound of the right hand side of Equation (6) in Lemma 3, i.e., $\bar{L}_{F,\mathcal{S}}^q \leq \tilde{L}_{F,\mathcal{S}}^q$.

## A.2 Redundant Constraints of the Lipschitz Model

In order to avoid possible numerical issues of problem (LipMON), and to improve the bounds, we add some redundant constraints to it. For variables $\mathbf{r}$ and $\mathbf{y}$. Note that $\mathbf{r} = (\mathbf{I}_p - \mathbf{W}^T \cdot \text{diag}(\mathbf{s}))^{-1} \cdot \mathbf{C}^T \mathbf{v}$, hence $\|\mathbf{r}\|_2 \leq \||(\mathbf{I}_p - \mathbf{W}^T \cdot \text{diag}(\mathbf{s}))^{-1}\||_2 \cdot \||\mathbf{C}^T\||_2 \cdot \|\mathbf{v}\|_2$. The operator norm of a matrix induced by $L_2$ norm is its largest singular value. Hence the operator norm of $(\mathbf{I}_p - \mathbf{W}^T \cdot \text{diag}(\mathbf{s}))^{-1}$ is the smallest singular value of matrix $\mathbf{I}_p - \mathbf{W}^T \cdot \text{diag}(\mathbf{s})$, which is smaller or equal than 1 from the recent work [46]. In summary, we have $\|\mathbf{r}\|_2 \leq \||\mathbf{C}\||_2 \cdot \|\mathbf{v}\|_2$ and $\|\mathbf{y}\|_2 \leq \||\mathbf{C}\||_2 \cdot \|\mathbf{v}\|_2$. For Lipschitz Model w.r.t. $L_2$ norm, we have $\|\mathbf{v}\|_2 \leq 1$; for Lipschitz Model w.r.t. $L_\infty$ norm, we have $\|\mathbf{v}\|_\infty^* = \|\mathbf{v}\|_1 \leq 1$, thus $\|\mathbf{v}\|_2 \leq \|\mathbf{v}\|_1 \leq 1$. Therefore, for both $L_2$ and $L_\infty$ norm, we can bound the $L_2$ norm of variables $\mathbf{r}$ and $\mathbf{y}$ by $\||\mathbf{C}\||_2$. Moreover, we multiply the equality constraint $\mathbf{r} - \mathbf{W}^T \cdot \mathbf{y} = \mathbf{C}^T \mathbf{v}$ coordinate-wisely with variables $\mathbf{s}, \mathbf{z}, \mathbf{y}, \mathbf{r}$ to produce redundant constraints and improve the results. This strengthening technique is already included in the software Gloptipoly3 [20]. With all the discussion above, we now write the strengthened version of problem (LipMON-b) as follows:

$$
\begin{aligned}
\max_{\mathbf{t},\mathbf{x}\in\mathbb{R}^{p_0},\mathbf{s},\mathbf{z},\mathbf{y},\mathbf{r}\in\mathbb{R}^p,\mathbf{v},\mathbf{w}\in\mathbb{R}^K} & \{\mathbf{t}^T\mathbf{U}^T\mathbf{y} : \|\mathbf{t}\|_q \leq 1, \ \mathbf{w}^T\mathbf{v} \leq 1, \ \|\mathbf{w}\|_q \leq 1, \ \mathbf{x} \in \mathcal{S}, \\
& \mathbf{s} \in \partial\text{ReLU}(\mathbf{Wz} + \mathbf{Ux} + \mathbf{u}), \ \mathbf{z} = \text{ReLU}(\mathbf{Wz} + \mathbf{Ux} + \mathbf{u}), \\
& \mathbf{r} - \mathbf{W}^T\mathbf{y} = \mathbf{C}^T\mathbf{v}, \ \mathbf{y} = \text{diag}(\mathbf{s}) \cdot \mathbf{r}, \ \|\mathbf{y}\|_2 \leq \||\mathbf{C}\||_2 \cdot \|\mathbf{v}\|_2, \ \|\mathbf{r}\|_2 \leq \||\mathbf{C}\||_2 \cdot \|\mathbf{v}\|_2, \\
& \mathbf{s}(\mathbf{r} - \mathbf{W}^T\mathbf{y}) = \mathbf{s}(\mathbf{C}^T\mathbf{v}), \ \mathbf{z}(\mathbf{r} - \mathbf{W}^T\mathbf{y}) = \mathbf{z}(\mathbf{C}^T\mathbf{v}), \\
& \mathbf{y}(\mathbf{r} - \mathbf{W}^T\mathbf{y}) = \mathbf{y}(\mathbf{C}^T\mathbf{v}), \ \mathbf{r}(\mathbf{r} - \mathbf{W}^T\mathbf{y}) = \mathbf{r}(\mathbf{C}^T\mathbf{v})\}. \quad\quad \text{(LipMON-c)}
\end{aligned}
$$

## A.3 Proof of Lemma 2

The SOS constraint in problem (EllipMON-SOS-$d$) can be written as

$$
\begin{aligned}
\sigma_0(\mathbf{x}, \mathbf{z}) = -\big( & \|\mathbf{Q}(\mathbf{Cz} + \mathbf{c}) + \mathbf{b}\|_2^2 - 1 \quad (=: f_1(\mathbf{x}, \mathbf{z})) \\
& + \sigma_1(\mathbf{x}, \mathbf{z})^T g_q(\mathbf{x} - \mathbf{x}_0) \quad (=: f_2(\mathbf{x}, \mathbf{z})) \\
& + \tau(\mathbf{x}, \mathbf{z})^T(\mathbf{z}(\mathbf{z} - \mathbf{Wz} - \mathbf{Ux} - \mathbf{u})) \quad (=: f_3(\mathbf{x}, \mathbf{z})) \\
& + \sigma_2(\mathbf{x}, \mathbf{z})^T(\mathbf{z} - \mathbf{Wz} - \mathbf{Ux} - \mathbf{u}) \quad (=: f_4(\mathbf{x}, \mathbf{z})) \\
& + \sigma_3(\mathbf{x}, \mathbf{z})^T\mathbf{z}) \quad (=: f_5(\mathbf{x}, \mathbf{z})) \\
= -\big( & f_1(\mathbf{x}, \mathbf{z}) + f_2(\mathbf{x}, \mathbf{z}) + f_3(\mathbf{x}, \mathbf{z}) + f_4(\mathbf{x}, \mathbf{z}) + f_5(\mathbf{x}, \mathbf{z})\big) =: -f(\mathbf{x}, \mathbf{z}).
\end{aligned}
$$

For $d = 1$, denote by $\mathbf{M}_i$ the Gram matrix of polynomial $f_i(\mathbf{x}, \mathbf{z})$ for $i = 1, \ldots, 5$ and $\mathbf{M}$ the Gram matrix of polynomial $f(\mathbf{x}, \mathbf{z})$, with basis $[\mathbf{x}^T, \mathbf{z}^T, 1]$. We have explicitly $\mathbf{M} = \sum_{i=1}^5 \mathbf{M}_i$, where $\mathbf{M}_i$

has the following form

$$
\mathbf{M}_1 = \begin{bmatrix} \mathbf{0}_{p_0 \times p_0} & \mathbf{0}_{p_0 \times p} & \mathbf{0}_{p_0 \times 1} \\ \mathbf{0}_{p \times p_0} & \mathbf{C}^T \mathbf{Q}^2 \mathbf{C} & \mathbf{C}^T \mathbf{Q}^2 \mathbf{c} + \mathbf{C}^T \mathbf{Q} \mathbf{b} \\ \mathbf{0}_{1 \times p_0} & \mathbf{c}^T \mathbf{Q}^2 \mathbf{C} + \mathbf{b}^T \mathbf{Q} \mathbf{C} & \mathbf{c}^T \mathbf{Q}^2 \mathbf{c} + 2\mathbf{b}^T \mathbf{Q} \mathbf{c} + \mathbf{b}^T \mathbf{b} - 1 \end{bmatrix},
$$

$$
\mathbf{M}_2 = \begin{cases} \begin{bmatrix} -\mathrm{diag}(\sigma_1) & \mathbf{0}_{p_0 \times p} & \mathrm{diag}(\sigma_1) \cdot \mathbf{x}_0 \\ \mathbf{0}_{p \times p_0} & \mathbf{0}_{p \times p} & \mathbf{0}_{p \times 1} \\ \mathbf{x}_0^T \cdot \mathrm{diag}(\sigma_1) & \mathbf{0}_{1 \times p} & \sigma_1^T(\varepsilon^2 - \mathbf{x}_0^2) \end{bmatrix}, & \text{for } L_\infty\text{-norm,} \\[2em] \sigma_1 \begin{bmatrix} -\mathbf{I}_{p_0} & \mathbf{0}_{p_0 \times p} & \mathbf{x}_0 \\ \mathbf{0}_{p \times p_0} & \mathbf{0}_{p \times p} & \mathbf{0}_{p \times 1} \\ \mathbf{x}_0^T & \mathbf{0}_{1 \times p} & \varepsilon^2 - \mathbf{x}_0^T \mathbf{x}_0 \end{bmatrix}, & \text{for } L_2\text{-norm,} \end{cases}
$$

$$
\mathbf{M}_3 = \begin{bmatrix} \mathbf{0}_{p_0 \times p_0} & -\frac{1}{2}\mathbf{U}^T \mathrm{diag}(\tau) & \mathbf{0}_{p_0 \times 1} \\ -\frac{1}{2}\mathrm{diag}(\tau)\mathbf{U} & \mathrm{diag}(\tau)(\mathbf{I}_p - \mathbf{W}) & -\frac{1}{2}\mathrm{diag}(\tau) \cdot \mathbf{u} \\ \mathbf{0}_{1 \times p_0} & -\frac{1}{2}\mathbf{u}^T \cdot \mathrm{diag}(\tau) & 0 \end{bmatrix},
$$

$$
\mathbf{M}_4 = \begin{bmatrix} \mathbf{0}_{p_0 \times p_0} & \mathbf{0}_{p_0 \times p} & -\frac{1}{2}\mathbf{U}^T \sigma_3 \\ \mathbf{0}_{p \times p_0} & \mathbf{0}_{p \times p} & \frac{1}{2}(\mathbf{I}_p - \mathbf{W}^T)\sigma_3 \\ -\frac{1}{2}\sigma_3^T \mathbf{U} & \frac{1}{2}\sigma_3^T(\mathbf{I}_p - \mathbf{W}) & -\sigma_3^T \mathbf{u} \end{bmatrix},
$$

$$
\mathbf{M}_5 = \begin{bmatrix} \mathbf{0}_{p_0 \times p_0} & \mathbf{0}_{p_0 \times p} & \mathbf{0}_{p_0 \times 1} \\ \mathbf{0}_{p \times p_0} & \mathbf{0}_{p \times p} & \frac{1}{2}\sigma_2 \\ \mathbf{0}_{1 \times p_0} & \frac{1}{2}\sigma_2^T & 0 \end{bmatrix}.
$$

Moreover, in order to improve the quality of the ellipsoid, we can also use the *slope restriction* condition of ReLU function as proposed in [22]: $(z_j - z_i)(\mathbf{W}_{j,:}\mathbf{z} + \mathbf{U}_{j,:}\mathbf{x} + u_j - \mathbf{W}_{i,:}\mathbf{z} - \mathbf{U}_{i,:}\mathbf{x} - u_i) - (z_j - z_i)^2 \geq 0$ for $i \neq j$. The Gram matrix of the SOS combination of these constraints with basis $[\mathbf{x}^T, \mathbf{z}^T, 1]$ has the form

$$
\mathbf{M}_6 = \begin{bmatrix} \mathbf{U} & \mathbf{W} & \mathbf{u} \\ \mathbf{0}_{p \times p_0} & \mathbf{I}_p & \mathbf{0}_{p \times 1} \\ \mathbf{0}_{1 \times p_0} & \mathbf{0}_{1 \times p} & 1 \end{bmatrix}^T \begin{bmatrix} \mathbf{0}_{p_0 \times p_0} & \mathbf{T} & \mathbf{0}_{p_0 \times 1} \\ \mathbf{T} & -2\mathbf{T} & \mathbf{0}_{p \times 1} \\ \mathbf{0}_{1 \times p_0} & \mathbf{0}_{1 \times p} & 0 \end{bmatrix} \begin{bmatrix} \mathbf{U} & \mathbf{W} & \mathbf{u} \\ \mathbf{0}_{p \times p_0} & \mathbf{I}_p & \mathbf{0}_{p \times 1} \\ \mathbf{0}_{1 \times p_0} & \mathbf{0}_{1 \times p} & 1 \end{bmatrix},
$$

where $\mathbf{T} = \sum_{i=1}^{p-1} \sum_{j=i+1}^{p} \lambda_{ij}(\mathbf{e}_i - \mathbf{e}_j)(\mathbf{e}_i - \mathbf{e}_j)^T$ with $\lambda_{ij} \geq 0$ for all $i < j$, and $\{\mathbf{e}_i\}_{i=1}^{p} \subseteq \mathbb{R}^p$ is the canonical basis of $\mathbb{R}^p$. Since $\sigma_0(\mathbf{x}, \mathbf{z})$ is an SOS polynomial of degree at most 2, we conclude that $-\mathbf{M} \succeq 0$. According to Lemma 5 in [14], the constraint $-\mathbf{M} \succeq 0$ is equivalent to an SDP constraint using *Schur complements*, which finishes the proof of Lemma 2.

## A.4 An Adversarial Example

## A.5 Licenses of Used Assets

Table 4: Summary of the licenses of used assets

| Software | License |
| --- | --- |
| Julia | MIT License |
| JuMP | Mozilla Public License |
| Matlab | Proprietary Software |
| CVX | CVX Standard License |
| Python | Python Software Foundation License |
| Pytorch | Berkeley Software Distribution |
| Mosek | Proprietary Software |
| Our code | CeCILL Free Software License |

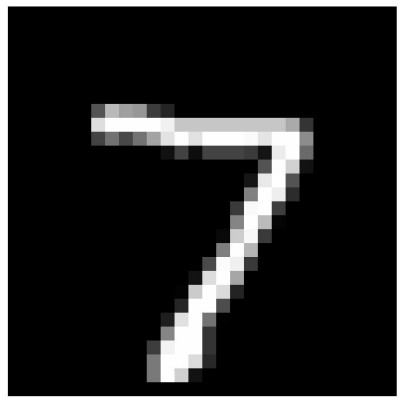 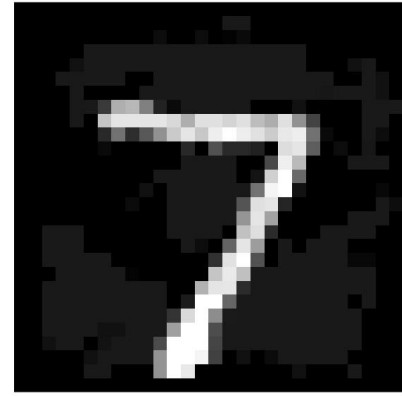

(a) Original example, classified as 7          (b) Adversarial example, classified as 3

Figure 2: An adversarial example of the first test MNIST input found by PGD algorithm for $L_\infty$ norm with $\varepsilon = 0.1$.