# OpenReview forum: "Semialgebraic Representation of Monotone Deep Equilibrium Models and Applications to Certification"
_NeurIPS.cc/2021/Conference — NeurIPS 2021 Poster_

### Official Review · Reviewer_UG93 · 2021-07-15

**Rating:** 6
**Confidence:** 3

**Summary:**

Summary:
The authors present three semialgebraic models that are applicable to fully connected monDEQs that are used to provide certificates of robustness of three different flavors. These are then applied to a monDEQ trained on MNIST and the resulting certificates are tighter than those generated by previous Lipschitz bounds.

**Limitations And Societal Impact:**

The limitations were discussed in the conclusion, but no attention was paid to the societal impacts of this work. Certification papers are quite benign, so it is unclear if such ethical discussion is required.

**Main Review:**

Originality:
While the formulation of robustness or lipschitz certificates in a semialgebraic model has been applied to deep neural networks in the past, such standard approaches have not yet been tried on monDEQs, which may be more amenable to these techniques given their smaller parameterizations. The ellipsoidal model is a novel form of reachability analysis to the best of my knowledge, and is possibly the most interesting novelty presented in this work. Several important works are not cited with respect to Lipschitz estimation: Jordan & Dimakis have several papers exactly computing or estimating Lipschitz constants (https://arxiv.org/abs/2003.01219, https://arxiv.org/abs/2107.02732), and these works (https://arxiv.org/pdf/2004.13135.pdf, https://arxiv.org/pdf/1908.06315v4.pdf ) discuss Lipschitz constants of implicit models.

Clarity:
The main contributions are clearly stated and the paper is fairly easy to read. The parallelism in the exposition of each of the three semialgebraic models is useful for cleanliness and clarity.

Quality:
The work is theoretically sound and the proofs I checked seem to be devoid of typos. Accepting that this work is primarily theoretical, the experiments are still quite limited. With no extra technical innovations, more experimental evaluations could be performed to examine: how performance/runtime are affected on monDEQs of varying #neurons or training regularizers/hyperparameters; the size of the safe-radii guaranteed by the Lipschitz bound; monDeqs trained on different datasets. Further, more traditional Lipschitz estimation techniques could be applied and compared against: e.g. CLEVER. Additionally, other POP-certification papers exploit sparsity to improve performance, but no mention of such approaches were discussed here: any attempts to improve runtimes would greatly strengthen this paper.

Significance:
The broader community has not yet latched on to monDEQs as a viable alternative for DNNs, so certification of these networks does not seem very useful for the community-at-large. On the other hand, the relatively small size of these nets can accelerate SDP-based certification schemes, and breakthroughs in this line of work could be a clear strength of DEQs over more traditional DNNs.

Overall:
This is a cute paper that uses reasonably standard techniques and only minor technical innovations to provide certifications on an interesting, but not-very-popular, class of models. The theory is sound and interesting, but the experiments are slightly limited.

Some typos/bugs:
- Some of the references in the appendix don't line up to the main bibliography. For example, ref 5 (line 526-527) in the appendix should not point to CNN-cert.
- There is a bug in the formulation of dual norms. Specifically, the second-to-third line in equation 8 is incorrect. Indeed, in the third line, w can be chosen to be zero and v can have unbounded norm, driving the objective value to infinity. Why not just keep the semialgebraic constraint that ||v||q*<= 1? (or alternatively flip the inequality constraining the q-norm of w)? This bug carries over to the main statement of Lemma 1.

**Time Spent Reviewing:**

5

---

> ### Author Response · Authors · 2021-08-05
> **Response to UG93:**
>
> ### Bibliography:
> We will cite the references proposed by the reviewer and add a discussion section in the appendix to present differences with related work.
>
> ### Quality:
> CLEVER would apply to implicitly defined input-output relations, but it is based on random sampling in a ball, and therefore does not provide certified bounds, contrary to the algebraic positivity certificate we use which allows to capture the entire ball. As for scalability, for example for convolutional networks, directly using off-the-shelves interior point SDP solvers for convolutional networks certification is not possible because of their current size limitation. Treating convolutional networks requires to take advantage of the sparsity of convolutions as well as possibly use specialized solvers (e.g. based on first-order methods as proposed in Dathathri et. al.). This is also mentioned by kdsN and ENDm, for example. We will expand on this in the main text.
>
> ### Typos:
> Thanks for pointing them out.
> We will of course correct them.

---

### Official Review · Reviewer_kbiJ · 2021-07-16

**Rating:** 6
**Confidence:** 3

**Summary:**

Based on semialgebraic of $\text{ReLU}$ and $\partial\text{ReLU}$, this paper proposes three models that are useful for certifying robustness, estimating Lipschitz constants and computing outer approximation ellipsoids.  The authors also use Putinar’s certificates to relax the non-convex problems to convex SDPs which can be solved efficiently. Experiments on MNIST demonstrate the effectiveness of all three models.

**Limitations And Societal Impact:**

Yes, the authors have adequately addressed the limitations and potential negative societal impact of their work.

**Main Review:**

[Strength]:

1 This paper is well-written and easy to read. The proofs are generally easy to follow (and pretty standard, to be honest).

2 The theoretical and empirical results demonstrate the improvement over the previous approach [24].

3 The empirical observation is interesting. ``DEQs are much less robust to $L^\infty$ perturbations than $L^2$ perturbations, in contrast with classical DNNs.”

[Weakness]:

1 While the results in this paper are interesting, I think that they are somewhat incremental. The idea of utilizing semialgebraic of $\text{ReLU}$ and $\partial\text{ReLU}$ is not new, even for monDEQs, see Proposition 5 in [c1], please. The approaches used are standard techniques. Moreover, as the authors point out, the size of the corresponding PSD matrix seems an unavoidable limiting factor. Thus, the proposed method appears troublesome to be extended to large models.

2 The relationship between the three models is unclear. In line 142, the authors say that “each model can be eventually used to certify robustness but they also have their own independent interest”. However, I do not find more description of “their own independent interest”. Did I miss something? There is a lack of discussion on the motivation of the three models.

3 The experiment is relatively weak. The empirical results are still on very small scales and not quite convincing. The authors conduct detailed comparison with [24], but the method are only evaluated on MNIST.

4 In line 43-45, “... DEQs have the definite advantage of being relatively small in size compared to DNNs and therefore...”. I do not agree with this argument. DEQs can be very large in size. As illustrated in previous empirical results, DEQs requires comparable parameters to achieve performance comparable to DNNs.

5 In Conclusion and Future work, the authors admit the limitation of the method. I really appreciate it. However, after reading your future work, I cannot feel the potential of this paper. I just feel that this paper is not well ready.

[Additional questions]

The authors observe that ``DEQs are much less robust to $L^\infty$ perturbations than $L^2$ perturbations, in contrast with classical DNNs.” To be honest, I think this experimental result is quite interesting. But is this only an empirical observation? Moreover, as I mentioned in Weakness 4, the experiments are on very small scales (MNIST), it seems that this observation is not quite convincing. I will appreciate it if you provide more explanations that are theoretical.

Overall, there is still a lot of room for improvement in this paper. I suggest weak acceptance for this paper, but I believe that with further refinement the authors could submit a better version to a future conference.

[c1] https://openreview.net/forum?id=bodgPrarPUJ.





**Time Spent Reviewing:**

48 hours

---

> ### Author Response · Authors · 2021-08-05
> **Response to kbiJ:**
>
> * 1: Proposition 5 in the cited reference is about the forward path of a ReLU monDEQ modeled as a convex optimization problem. It is based on semialgebraicity of the ReLU function, but it is used for a purpose very different from ours.
>
> * 2: We will clarify the interest of each model. The purpose of this sentence is to highlight the fact that robustness certification is a common task which can be tackled using each model respectively. However the respective outputs are different, which  justifies a comparison of these models on this task. On the other hand, the ellipsoid model is typically used for reachability analysis (as pointed out by reviewer UG93) while the robustness certification model cannot be used for this purpose for example.
>
> * 3: As for scalability, for example for convolutional networks, directly using off-the-shelves interior point SDP solvers for convolutional networks certification is not possible because of their current size limitation. Treating convolutional networks requires to take advantage of the sparsity of convolutions as well as possibly use specialized solvers (e.g. based on first-order methods as proposed in Dathathri et. al.). This is also mentioned by kdsN and ENDm, for example. We will expand on this in the main text.
>
> * 4: The purpose of this sentence was to highlight the fact that DEQs are simpler in the sense that there is usually a single hidden layer. This is a structural advantage for certification provided that we can handle the implicit nature of the input output relation, which is what we propose in this work.
>
> * 5: The main purpose of the paper is to show that SDP-based certification is applicable to implicitly defined networks. As for usual deep networks, the main challenge is to find an acceptable compromise between scalability and performance, e.g. with dedicated solvers (see response to kdsN and ENDm about Dathathri et al). This paper constitutes a first step, which is indeed limited to modest size monDEQs, as was done in reference [28] for usual feed forward networks. This opens a research perspective regarding scalability, for which advances require significant work. Also, as another contribution, we have introduced a reachability analysis tool for monDEQs with the ellipsoid, as pointed out by UG93.
>
> * Additional question: We do not have a theoretical explanation for this observation. As explained in the response to SqFE, fully connected feedforward networks have been shown to be robust to the level we are considering (epsilon = 0.1) on MNIST in reference [28]. Our only intuition is that DEQ layers act as a very large number of layers so that some robustness is lost as the network capacity is much broader. This remains only an intuition and would require further analysis to make it a theoretical claim.

---

> ### Comment · Area_Chair_2MMz · 2021-08-20
> **Does the author response address your concerns?**
>
> Dear Reviewer,
> I would encourage you to read the author rebuttal and give your further comments. Please pay attention to whether you want the authors to read your further comments. Thanks!
>
> Area Chair

---

> > ### Comment · Reviewer_kbiJ · 2021-08-22
> > **Good paper**
> >
> > The rebuttal has addressed my concerns. After reading other reviews and rebuttals, I'd like to see this paper accepted.

---

### Official Review · Reviewer_ENDm · 2021-07-16

**Rating:** 6
**Confidence:** 3

**Summary:**

The authors propose a framework for the robustness verification of monDEQs (monotone operator deep equilibrium networks). In this framework the paper introduces three different methods: one for direct robustness verification, one for Lipschitz constant estimation and one to compute the ellipsoid over-approximating all possible outputs for a set of inputs (in a norm ball).
The authors also show how the Lipschitz constant and ellipsoid in turn can be used for robustness verification.

Mathematically this is enabled encoding the monDEQ as the constraints of a (polynomial) optimization problem.
In particular, the fix-point $z = ReLU(Wz + Ux_{0} + u)$ is encoded using the semialgebracity of ReLU, similar to [27]*. This encoding as well as encoding $\ell_{2}$ and $\ell_{\infty}$ inputs yields quadratic constraints. (Other $\ell_{p}$-norms $p>2$ yield higher-order constraints.)
The resulting constraints are relaxed via Puntinar's positivity certificate and Lasserre's/Shor's relaxations into a SDP (semidefinite program), e.g. approximated lower-degree polynomials.

For the robustness verification approach the authors follow [27] and maximize the difference between any logit and the target logit. If all of these differences are $< 0$ a robustness certificate is obtained. To make the resulting problem tractable Shor's relaxation is applied.
The Lipschitz approach bounds the Lipschitz constant by bounding the supremum of the operator norm of the generalized Jacobian. This in turn, via Shor's relaxation gives an SDP.
The Ellipsoid model minimizes the volume of an ellipsoid that covers the output. Via a custom relaxation this also yields an SDP.

All methods are evaluated on an MNIST classifier with a single feed-forward fix-point layer where $z \in \mathbb{R}^{87}$.
All three approaches allow for robustness certification (although the robustness formulation performs the best).
The computed Lipschitz bound are close to those obtained by [24] for the $\ell_{2}$-norm but much tighter for the $\ell_{\infty}$-norm. Lastly, in an example the authors showcase how the output-ellipsoid method allows insights into the model robustness.


**Limitations And Societal Impact:**

The authors discuss technical limitations and note that due to the theoretical nature of the work no direct negative social impacts can be foreseen.
This seems sufficient for this kind of work.


**Main Review:**

Overall the paper is well written and appears to be technically sound.
While well presented, the overall tone is very technical tone and readers might benefit from deferring technical details to the appendix in favor of more intuitive explanations (e.g. L112-L136).

The mathematical arguments in the main text seem correct. However, I did not check the proof of Lemmas 3 in detail. Based on it the construction of Lemma 1 it seems correct and so od the constructions in Section 3.3 and Lemma 2.

The main drawback of the approach appears to the scalablity of the SDP solver as well as the fact that the approach currently is tied to single-layer ReLU feed-forward models.
In comparison, [24] is much more scalable and less limited in architecture. Yet, their approach is limited to $\ell_{2}$ Lipschitz constants.

While the paper might benefit from a wider discussion of related certification approaches, the most closely related works (with one notable exception -- see below) seem to be cited.

Questions:
- It seems the related work is missing a discussion of [Dathathri et al.]. In particular, can you comment on wether their solver would allow you to certify larger monDEQs as you name the underlying SDP solver as a key limitation in L252-254,L347.
- In 3.2 you introduce the Lipschitz constant w.r.t the set $\mathcal{S}$ and in L296 you say that you compute global Lipschitz constants. So you take $\mathcal{S}$ as the set of all legal inputs?
- The choice of specifically 87 neurons was made due to the capability of the SDP solver?
- I am unsure why convolutional layers can not be handled, as they (as you point out in L351-352) can be encoded as linear layers. Is this again an issue of size?
- While definitely intresting on a technical level, can you speak to possible uses of the output ellipsoid? It seems to be well suited to (multi-dimensional) regression tasks. How would it compare to a (local) Lipschitz constant in such cases.

References:

Dathathri et al., Enabling certification of verification-agnostic networks via memory-efficient semidefinite programming, NeurIPS'20


**Time Spent Reviewing:**

10

---

> ### Author Response · Authors · 2021-08-05
> **Response to ENDm:**
>
>
>
>
> * Dathathri et al.: We will include a discussion of this paper as it is an important contribution which our text has missed. The idea of having an SDP solver based on available computational oracles (forward and backward propagation) adapted to neural network certification is very relevant to our work too. We believe that such techniques could be applied to our problem, although this would require a non trivial adaptation. Indeed Dathathri et. al. use the forward and backward propagation as algorithmic oracles. In the context of implicitly defined layers this should be adapted, as forward propagation is fixed point solving, and backward propagation is implicit differentiation (in particular intermediate steps do not need to be stored in memory). This is an important topic of future research for our work, but would require a significant extension.
>
> * We will clarify the global Lipschitz constant issue. In this case, S is a large ball which contains all the input points. Such an approach was used in reference [7] for example.
> * As for the 87 neurons, we repeated the experiment in reference [25] based on the proposed code. We did not change the number and it turned out to be moderate enough to fit our SDP solver.
> * Directly using off-the-shelves interior point SDP solvers for convolutional networks certification is not possible because of their current size limitation. Treating convolutional networks requires to take advantage of the sparsity of convolutions as well as possibly use specialized solvers (e.g. based on first-order methods as proposed in Dathathri et. al.). This is also mentioned by kdsN, for example. We will expand on this in the main text.
>
> * We will expand on potential use of the output ellipsoid. One important purpose of this model is for reachability analysis as pointed out by UG93.

---

> > ### Comment · Reviewer_ENDm · 2021-08-31
> > **Thank you**
> >
> > Dear Authors,
> >
> > Thank you for reply.
> > Having read the other reviews as well as your responses I don't see any major concerns and retain my initial score.
> >
> > Best,
> > Reviewer ENDm

---

> ### Comment · Area_Chair_2MMz · 2021-08-20
> **Does the author response address your concerns?**
>
> Dear Reviewer,
> I would encourage you to read the author rebuttal and give your further comments. Please pay attention to whether you want the authors to read your further comments. Thanks!
>
> Area Chair

---

### Official Review · Reviewer_kdsN · 2021-07-16

**Rating:** 7
**Confidence:** 5

**Summary:**

Authors consider the certification of properties of deep equilibrium models. Some of these properties are novel and the mathematical apparatus employed is very elegant, if not completely novel.



**Ethical Concerns:**

This could also give false hope of being able to certify properties of real-world applications of neural networks to mission-critical applications, which we are very far from.

**Limitations And Societal Impact:**

This could help improve the applicability of neural networks to mission-critical applications, eventually.

**Main Review:**

Originality:

The authors consider the problem of verification of properties of monotone deep equilibrium models [1], while using techniques of polynomial optimization [2,3], or rather the first level of several hierarchies (wherein their coincide and which is known as Shor's relaxation). While both the problem and the use of polynomial optimization therein is relatively recent, the extent to which the present paper is "original" is much less than [1,2,3]. Specifically, the semialgebraic form or ReLU (line 109) and Putinar's positivity certificate (line 112 and following) are well known and the inequalities for testing Lipschitz constants (line 176) have been used previously, as suggested by the authors themseleves. So the contribution lies in
-- generalizing the results related to Lipschitz constants to arbitrary L_q norms in Lemma 1
-- utilizing a well-known "minimum-volume enclosing ellipsoid" technique in Lemma 2, for the first time in verification of neural networks.
This contribution is not particularly well articulated.

Quality:

Still, the problem of verification of properties of deep-learning approaches is of considerable interest, and the authors make several contributions. Crucially, these may enable further work, such as the use of the Frank-Wolfe methods in the ellipsoid model [5]

Computationally, the authors find instances where the generalized SemiUB_f^q provides substantially better (cca. 8x) bounds than previous works. In Figure 1, they provide a very impressive example of the use of the ellipsoid model. Throughout, as can be expected, the computational results are limited to small-scale, hand-picked illustrations.

It may be worth pointing out that the contribution of the paper is essentially conceptual, rather than presenting a complete system or even a scalable approach. Other systems and approaches (not cited by the present authors), e.g.:
https://arxiv.org/abs/2002.06864
https://arxiv.org/abs/2007.10868
https://arxiv.org/abs/2103.03638
routinely scale to millions of neurons, which may be impossible even with Shor's relaxation.

Clarity:

The text is not particularly well structured. For instance, Section 4.3 explains again the ellipsoid model, which has been introduced twice previously?

The discussion of the empirical results of Section 4.1 is hard to relate to the results of Table 2.

The authors do not cite important recent drafts, such as [4].

Some of the sentences read a bit like Google Translate from some language other than English:
"The training of DEQs involves solving fix-point equations for which algorithmic success requires conditions."

--
[1] Chirag Pabbaraju, Ezra Winston, and J. Zico Kolter. Estimating lipschitz constants of monotone deep equilibrium models. In International Conference on Learning Representations, 2021.

[2] Tong Chen, Jean B Lasserre, Victor Magron, and Edouard Pauwels. Semialgebraic optimization for lipschitz constants of relu networks. In H. Larochelle, M. Ranzato, R. Hadsell, M. F. Balcan, and H. Lin, editors, Advances in Neural Information Processing Systems, volume 33, pages
375 19189–19200. Curran Associates, Inc., 2020.

[3] Mahyar Fazlyab, Alexander Robey, Hamed Hassani, Manfred Morari, and George Pappas. Efficient and accurate estimation of lipschitz constants for deep neural networks. In H. Wallach, 384 H. Larochelle, A. Beygelzimer, F. d'Alché-Buc, E. Fox, and R. Garnett, editors, Advances in Neural Information Processing Systems, volume 32. Curran Associates, Inc., 2019.

[4] Sumanth Dathathri, Krishnamurthy Dvijotham, Alexey Kurakin, Aditi Raghunathan, Jonathan Uesato, Rudy Bunel, Shreya Shankar, Jacob Steinhardt, Ian Goodfellow, Percy Liang, Pushmeet Kohli: Enabling certification of verification-agnostic networks via memory-efficient semidefinite programming. https://arxiv.org/abs/2010.11645

[5] S Damla Ahipasaoglu, P Sun, MJ Todd: Linear convergence of a modified Frank–Wolfe algorithm for computing minimum-volume enclosing ellipsoids. Optimisation Methods and Software 23 (1), 5-19 (2008).


**Time Spent Reviewing:**

4

---

> ### Author Response · Authors · 2021-08-05
> **Response to kdsN:**
>
> ### Originality
> Indeed, the main originality is to successfully apply the proposed techniques to monDEQ models, this was not proposed before. We will clarify this in the main text. One of the messages of the paper is that monDEQs have a favorable structure for certification because they usually contain a single hidden layer.
>
> ### Quality
> Indeed, extension to more complicated networks, such as convolutional networks, requires to take advantage of the sparsity of convolutions as well as possibly use specialized solvers (e.g. based on first-order (subgradient) methods as proposed in Dathathri et. al.). This is also mentioned by ENDm, for example. We will expand on this in the main text.
>
> ### Clarity
> We will work on the presentation and clarity and in particular add the suggested reference [4] as it is an important element of discussion (see the response to ENDm). We will also discuss more thoroughly the different purpose of each model, and avoid repetitions by using more pointers to previous sections.

---

### Official Review · Reviewer_SqFE · 2021-07-28

**Rating:** 6
**Confidence:** 4

**Summary:**

This paper introduces a semialgebraic representation of the monotone deep equilibrium model (monDEQ), a type of implicit-depth equilibrium network which guarantees the convergence of fixed-point iterations. This representation is used to derive SDP models for 1) certifying robustness, 2) estimating the Lipschitz constant, and 3) ellipsoid uncertainty propagation. These techniques show improved performance over existing methods for certifying robustness and bounding the Lipschitz constant of monDEQs. In addition, they apply to general $\ell_p$ norms, unlike existing work.


**Limitations And Societal Impact:**

Yes

**Main Review:**

### Pros
* This paper deals with an interesting topic which is gaining in importance as there is increasing attention on implicit-depth models. To date, there is very little work dealing with the robustness of implicit-depth models, and techniques that apply to standard DNNs cannot be applied.
* The paper is quite well written and thoroughly presented.

### Cons
* The LBEN implicit network of [Revey et al.](https://arxiv.org/pdf/2010.01732.pdf) is not cited or compared to. It generalizes the monDEQ as well as admitting a Lipschitz bound during training.
* The experimental results are very small-scale. It would be nice to see larger-scale experiments, for example on CIFAR-10. Does the computational complexity of the SDPs rule this out?


Additional comments/questions:
 * I think that the characterization of monDEQ as "less robust" to $\ell_\infty$ attacks than to $\ell_2$ attacks is a bit misleading. Isn't it common to to use attack radii of different scales when comparing attacks of different norms?  See for instance Appendix H  in [Wong et al. 2018](https://arxiv.org/pdf/1805.12514.pdf), where they scale $\ell_2$ attacks up a lot so as to have comparable volume of the attack regions across norms.
 * In two places you state that the network is robust _if and only if_ the projection $\mathcal{P}_i(\mathcal{C})$ lies below the classification boundary (lines 245 and 322). I'm confused where the "only if" comes from?
 * For the Lipschitz model, why was $\mathcal{S}$ chosen to be a single ball around all the test points, rather than an individual ball for each test point to be certified. Was this just for computational reasons?




**Time Spent Reviewing:**

4

---

> ### Author Response · Authors · 2021-08-05
> **Response to SqFE:**
>
> ### LBEN of Revey et. al.
> The purpose of this work is to enforce Lipschicity during training, which is different from our purpose. The approach is close to that of Reference [25] which is more recent: a special parametrization ensure Lipschicity with a given constant. On the other hand, we certify a posteriori monotone equilibrium networks without any assumption regarding the way they are trained. Furthermore, in [25], estimating a posteriori the Lipschitz constant of a monDEQ amounts to solve an eigenvalue problem, whereas for LBEN, it requires to solve a Bilinear Matrix inequality (BMI) plus a bisection, which is very costly. Therefore we stick to [25] for comparison purpose. We will cite the mentioned reference and discuss our results in light of this reference and the above comment. This was also pointed out by kbiJ.
>
> ### Small scale experiments
> Directly using off-the-shelves interior point SDP solvers for convolutional networks certification is not possible because of their current size limitation. Treating convolutional networks requires to take advantage of the sparsity of convolutions as well as possibly use specialized solvers (e.g. based on first-order methods as proposed in Dathathri et. al.). This is also mentioned by kdsN and ENDm, for example. We will expand on this in the main text.
>
> ### Additional questions
>
> * The comment on L infinity robustness refers also to the fact that fully connected feedforward networks have been shown to be robust to the level we are considering (epsilon = 0.1) on MNIST in reference [28]. We will clarify our statement.
> * Thanks for pointing out the only if issue, the reviewer is right, this will be corrected.
> * Choosing a unique ball for all points requires to solve one SDP problem, while one ball for each point (say among N) requires to solve N SDPs which is much more costly. In this case it is more favorable to use the adversarial robustness model directly.

---

> ### Comment · Area_Chair_2MMz · 2021-08-20
> **Does the author response address your concerns?**
>
> Dear Reviewer,
> I would encourage you to read the author rebuttal and give your further comments. Please pay attention to whether you want the authors to read your further comments. Thanks!
>
> Area Chair

---

### Decision · Program_Chairs · 2021-09-27

**Decision:**

Accept (Poster)

**Comment:**

The paper got 4 "Marginally above the acceptance threshold"s and 1 "Good paper, accept", all with relatively high confidences. The author rebuttals mostly addressed the concerns from the reviewers. Reviewers UG93, kdsN, SqFE, and ENDm were all in favor of accepting the paper in their post-rebuttal opinions. Thus the AC recommended acceptance. Nonetheless, some issues remain, e.g., scalability of the experiments, a bit overselling, etc. The AC also found that the "certification" problem, the core topic of the paper, was not clearly defined (lines 29-30 may allude to, but still in a vague manner). Hope these issues could be addressed in the revision.